# The effect of adherence to spectacle wear on early developing literacy: a longitudinal study based in a large multiethnic city, Bradford, UK

Alison Bruce,[1] Brian Kelly,[1] Bette Chambers,[2] Brendan T Barrett,[3] Marina Bloj,[3] John Bradbury,[4] Trevor A Sheldon[5]

[1]Bradford Institute for Health Research, Bradford Teaching Hospitals NHS Trust, Bradford, UK
[2]Institute for Effective Education, University of York, York, UK
[3]School of Optometry and Vision Science, University of Bradford, Bradford, UK
[4]Bradford Teaching Hospitals NHS Foundation Trust, Bradford, UK
[5]Department of Health Sciences, University of York, York, UK

**Correspondence to**
Dr Alison Bruce;
alison.bruce@bthft.nhs.uk

## ABSTRACT

**Objectives** To determine the impact of adherence to spectacle wear on visual acuity (VA) and developing literacy following vision screening at age 4–5 years.

**Design** Longitudinal study nested within the Born in Bradford birth cohort.

**Setting and participants** Observation of 944 children: 432 had failed vision screening and were referred (treatment group) and 512 randomly selected (comparison group) who had passed (<0.20 logarithm of the minimum angle of resolution (logMAR) in both eyes). Spectacle wear was observed in school for 2 years following screening and classified as adherent (wearing spectacles at each assessment) or non-adherent.

**Main outcome measures** Annual measures of VA using a crowded logMAR test. Literacy was measured by Woodcock Reading Mastery Tests-Revised subtest: letter identification.

**Results** The VA of all children improved with increasing age, −0.009 log units per month (95% CI −0.011 to −0.007) (worse eye). The VA of the adherent group improved significantly more than the comparison group, by an additional −0.008 log units per month (95% CI −0.009 to −0.007) (worse eye) and −0.004 log units per month (95% CI −0.005 to −0.003) in the better eye. Literacy was associated with the VA, letter identification (ID) reduced by −0.9 (95% CI −1.15 to −0.64) for every one line (0.10 logMAR) fall in VA (better eye). This association remained after adjustment for socioeconomic and demographic factors (−0.33, 95% CI −0.54 to −0.12). The adherent group consistently demonstrated higher letter-ID scores compared with the non-adherent group, with the greatest effect size (0.11) in year 3.

**Conclusions** Early literacy is associated with the level of VA; children who adhere to spectacle wear improve their VA and also have the potential to improve literacy. Our results suggest failure to adhere to spectacle wear has implications for the child's vision and education.

## INTRODUCTION

Visual development in humans occurs in early life[1] with the presence of reduced visual acuity (VA) in young children potentially indicating an associated condition such as significant refractive error, strabismus and/or amblyopia.[2] The UK National Screening Committee (UK NSC) recommends visual screening for all children at age 4–5 years,[3] (first year of school) in order to identify a potential reduction in VA. For those who fail the screening test (>0.20 logarithm of the minimum angle of resolution (logMAR) in one or both eyes),[3] the follow-up clinical pathway includes referral for a cycloplegic refraction and fundus examination to confirm the VA finding, to determine the presence and magnitude of any refractive error and to rule out eye disease.[4] In those with reduced VA, treatment generally consists of the wearing of spectacles[5] and may be combined with occlusion therapy[6] (wearing an eye patch or atropine drops). However, adherence to treatment, both spectacle wear[7 8] and occlusion therapy, is known to be variable.[9]

Decreased VA, both near and distance and also the presence of refractive error in young children, has been reported to be associated with reduced literacy levels.[10–12] However, there is a paucity of evidence on the impact

of non-adherence to spectacle wear on VA and early developing literacy in children. Early literacy skills such as letter recognition,[13] word reading and decoding[14] taught in the first years of school are indicators of future reading performance and educational attainment, which in turn affect long-term health and social outcomes.[15 16] The initial school years are a crucial time for the development of these key literacy skills[17] and it is important to understand the impact of non-adherence to spectacle wear on visual outcome and educational attainment.

Low educational attainment is associated with socioeconomic deprivation,[16] which makes the investigation of the relationship between VA and literacy difficult, as in order to account for potential confounding factors, comprehensive epidemiological data are required. Born in Bradford (BiB) is a large birth cohort, which collected maternal and early-life measures from mothers and their children in Bradford and details of recruitment have been previously reported.[18] By linking separately collected vision and literacy data in children in the BiB cohort, we had the opportunity to explore the association between VA, spectacle wear and literacy development while taking into account the effects of potential confounders. The aim of this study is to examine the impact of adherence to spectacle wear on VA and early developing literacy skills in children during their first 3 years of school.

## METHODS
This is a prospective, longitudinal study nested within the BiB cohort following children from the point of their initial vision screening at age 4–5 years. The study took place between 2012 and 2015. Baseline epidemiological data collected from mothers and children of the BiB cohort, literacy measures, vision screening results and repeat measures of vision and literacy were linked in order to evaluate the longitudinal impact of adherence to spectacle wear on VA and early literacy.

## Population
All children invited to join the study were participating in the BiB,[18] a longitudinal, multiethnic birth cohort study aiming to examine the impact of environmental, psychological and genetic factors on maternal and child health and well-being. Bradford is an ethnically diverse city (approximately, half of the births are to mothers of South Asian origin) with high levels of socioeconomic deprivation. The cohort is broadly representative of the city's maternal population of childbearing age.

## Patient and public involvement
The BiB project emphasises the importance of involving parents and ensuring they are central to the research that is prioritised; what is important to the parents, how people find out the results from the research projects and what it means for their families. The participants were asked their views on many research topics including literacy levels, vision and the impact of vision on literacy.

The participants suggested that these topics are of high importance and should be prioritised. The preliminary findings have been reported to the parents to provide verification of the data, ensuring that the findings reflect true patient experiences. Their ideas are essential in developing and revising current information provided to parents and carers. Their involvement has allowed the research to be prioritised around the needs and requirements of patients and carers. Finally in the dissemination of the research results, the parents will be central to publicising this study and its findings to local people, schools and the wider community.

## Recruitment
As part of a BiB study, children's literacy levels on school entry (termed 'Reception Class' in England, UK and defined as year 1 of this study) were measured between September 2012 and July 2014 in Bradford schools. Two thousand nine hundred and thirty BiB children from 74 of the 123 primary schools (60%) participated. Of the 2930, 432 (14.7%) failed their vision screening (figure 1) and were referred for follow-up cycloplegic investigation, these children are defined as the treatment group. A further 512 BiB children from the same schools (randomly selected using Excel's random number generator) who had passed vision screening were also invited to participate and were defined as the comparison group, giving a total of 944 participants in the study. Consent was opt-out and parents received a letter via the schools requesting continued participation prior to each annual assessment. Of the 944, 893 (94.6%) consented to participate in year 2 and 650/944 (68.9%) participated in year 3 (figure 1).

## Baseline vision assessments: year 1
The vision screening programme for children aged 4–5 years in Bradford is conducted in the first year of school by orthoptists with 97% of eligible children being screened.[19] The screening includes standard protocols for measurement of monocular distance VA.[20 21] VA was measured at a distance of 3 m using the LogMAR Crowded Test (Keeler, Windsor, UK) which has four letters per line, with each letter having a score of 0.025; the total score for each line thus represents 0.10 log unit (online supplementary information 1). A matching card was used and knowledge of letters was not therefore necessary to perform the test. VA was measured to threshold (ie, best achievable VA with no defined endpoint). In addition, cover test at 6 m and 1/3 m was performed. The data formed the baseline vision data (year 1). No child in the study was wearing spectacles at the baseline assessment.

Children failing to achieve the VA pass criterion (>0.20 logMAR in one or both eyes) set by the UK NSC[3] or who had a strabismus detected on cover testing were referred for follow-up. The standard clinical pathway[4] following vision screening entailed referral to either to a community optometrist or the hospital eye service where a cycloplegic refraction (1% cyclopentolate hydrochloride)

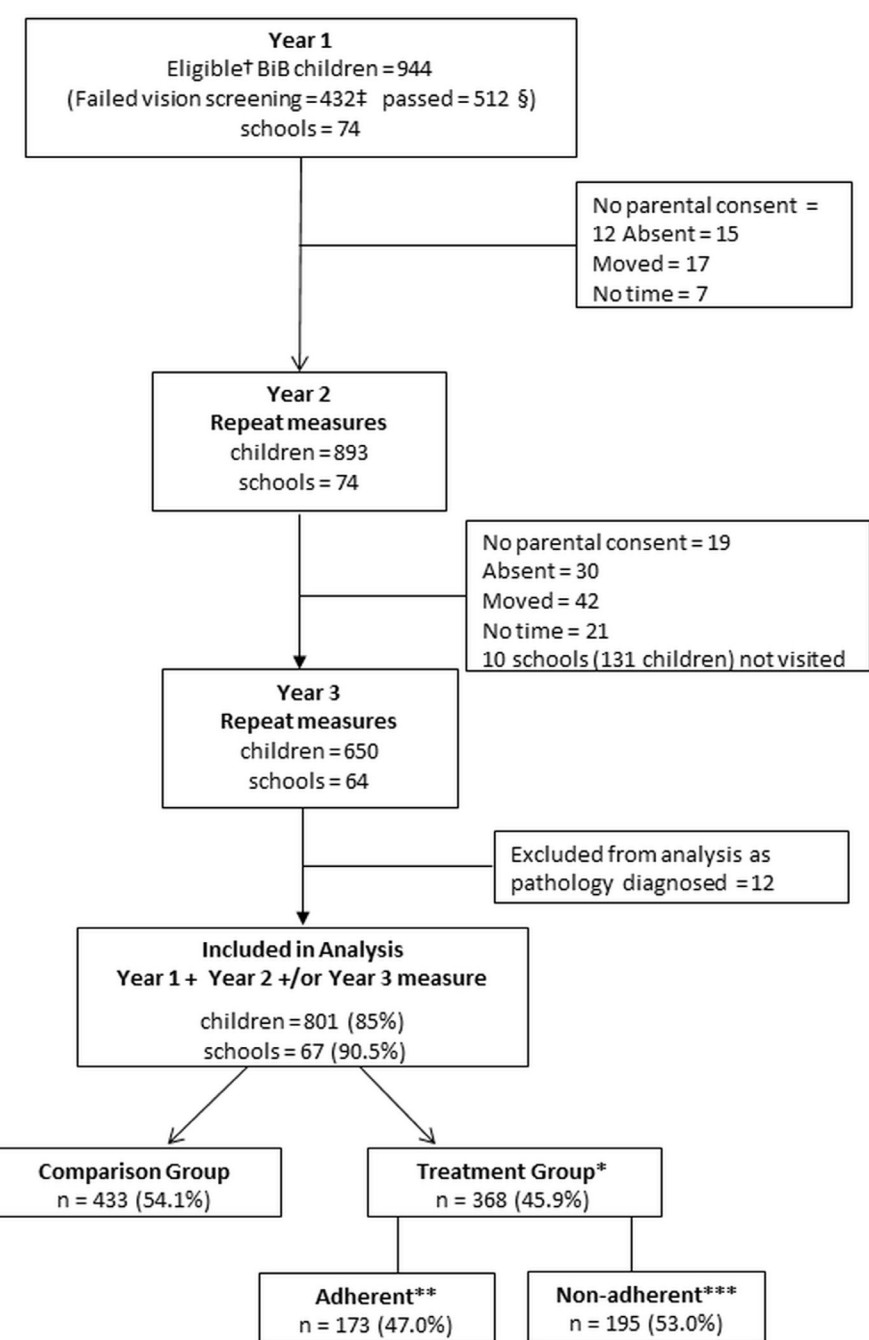

**Figure 1** Flow chart of the study participants. *Treatment group=children who failed vision screening and were referred for cycloplegic assessment. **Adherent=prescribed spectacles worn at each visual acuity assessment. ***Non-adherent=children who failed to attend cycloplegic examination and also children who attended but failed to wear prescribed spectacles at each visual acuity assessment. †Total number of eligible BiB children. ‡All BiB children who failed vision screening and additionally had a literacy score measured during the same school term. §Random sample of BiB children who passed vision screening and additionally had a literacy score measured during the same school term. BiB, Born in Bradford.

and fundus examination were undertaken, either by a paediatric ophthalmologist or an optometrist. Spectacles were prescribed based on the result of the cycloplegic refraction and clinical judgement; children were generally prescribed spectacles, including low degrees of hypermetropia (>+1.00 DS to +3.00 DS), if they had a reduced VA. A follow-up appointment was then arranged with the orthoptist approximately 8 weeks after the cycloplegic examination to repeat the VA measurement, with

the child wearing spectacles if they had been prescribed. Children assessed by a community optometrist of their choice had the results of their examination returned to the hospital eye service and also had a follow-up appointment arranged with an orthoptist.

All VA testing, both at the point of vision screening and at follow-up, was performed using the same method of measurement. The results of the follow-up assessment, including cycloplegic refraction, VA with the prescribed

glasses, cover testing and fundus and media examination, were extracted from the medical notes. The ophthalmic staff did not have knowledge of the baseline literacy assessment.

## Baseline literacy assessments: year 1

Literacy was measured on school entry (year 1) by trained research assistants within the same academic term as the vision screening. The research assistants were unaware of the VA results. An age-appropriate literacy measure, the Woodcock Reading Mastery Tests-Revised subtest: letter identification (ID), a validated reading skill test, was used to assess early literacy.[22] Letter-ID measures the child's ability to identify single letters, an essential skill mastered prior to reading and one of the best predictors of future reading achievement.[15] The letter-ID test is a test of knowledge of letters (the complete alphabet is used) and the child must verbally identify the name of each letter. This literacy measure specifically uses varied font type; the size of the letters approximate to 1.1 log unit (20/250) at 33 cm, therefore, the performance on this test is not affected by the level of VA. Letter-ID was collected in both raw and age-standardised format. In addition, receptive vocabulary was measured using the British Picture Vocabulary Scale (BPVS)[23] an indicator of cognitive ability, providing a representation of IQ in young children. This measure is included to adjust for potential confounding due to levels of general cognitive ability.

## Follow-up assessments: years 2 and 3

Vision and literacy measures were repeated within the same school term approximately 12 months (year 2) and 24 months (year 3) after the baseline measurements. Both the vision and the literacy assessments were administered on the same day by the same personnel who were unaware of previous vision or literacy results. VA and literacy was measured as detailed above. VA found to be ≥0.10 logMAR was repeated with a pinhole and near VA was measured using the Bailey-Lovie near-vision chart[24] (online supplementary information 1) and whether the child was wearing spectacles was recorded. In order to present the real-life impact of adherence to spectacle wear, all VA measures reported are presenting VAs, that is, measured with spectacles if worn at the time of the assessment in school. Parents and children were not given prior warning of these assessments.

## Statistical analysis

Children with baseline data for both vision and literacy in year 1 and who had at least one follow-up measure in either year 2 or year 3 were included in the final analysis (figure 1). The statistical model selected for the analyses, using projections over time, takes into account missing data and requires a minimum of measures at two time points. Using this type of statistical analysis allows inclusion of a greater number of participants giving maximum power to the analyses.[25] The characteristics of children participating in the study were compared initially using

$\chi^2$ test or two-sided t-tests as appropriate. Children in the treatment group were retrospectively divided into two subgroups, adherent and non-adherent. Adherence was defined as wearing prescribed spectacles at the time of assessment; otherwise, children were defined as non-adherent. Children who were assessed two times but only wore the spectacles on one occasion were classed as non-adherent. A sensitivity analysis was conducted to assess the extent to which the results varied by changing the definition of adherence.

### Analysis of VA

To investigate the effect of spectacle wear over time on VA, multilevel longitudinal models[25] were first constructed with VA as the outcome measure for the child's better and worse eye. The models measure change within the individual and change between individuals over time and allow for individual differences in the rate of change over time.[25] A quadratic term was included to model the non-linear trajectory of change. The model also includes an interaction term to compare the relationship between age and group, to test whether differences by group are the same at different ages. Unadjusted analysis was initially undertaken with subsequent adjustment for demographic and socioeconomic factors reported in the literature to be associated with reduced VA: early-life factors[26] (gender, gestational age, birth weight, route of birth) and maternal factors[27] (ethnicity, mother's age at delivery, mother's level of educational attainment and being in receipt of means-tested benefits). Predicted outcomes were plotted to visualise group differences and change in the outcomes for each group over time.

### Analysis of literacy

In order to estimate the association between the letter-ID and VA, the same multilevel and longitudinal modelling approach was adopted, but with the final letter-ID score as the outcome measure. The raw letter-ID scores were used in the analysis in order to explore change over time. After estimating differences between the groups and accounting for the initial letter-ID at baseline (year 1), further adjustment was undertaken for the factors reported in the literature to be associated with educational attainment,[28 29] the early-life factors and maternal factors as stated above. Spherical equivalent refraction (SER) (sphere plus half cylinder) of the better eye was included as was BPVS score in order to account for cognitive ability. The results of these models are presented along with predicted outcomes for each of the groups. Effect sizes are generally reported when appraising educational interventions. To demonstrate group differences at each time point, effect sizes were calculated for the letter-ID scores using Cohen's d.[30]

### VA: year 3

Children were unable to accurately perform the near VA (logMAR) test until year 3; we are therefore unable to provide a longitudinal analysis. In year 3, we have

measures of both near VA and distance VA and present the correlation between the near and distance VA at this time point only. Additionally, we analysed the association between near VA and literacy to examine if the results differed from the association between distance VA and literacy in year 3 only.

All analyses were carried out using Stata V.13 (StataCorp).

## RESULTS

Data from 801 (85%) children from 67 schools were included in the final analysis (figure 1). Twelve children in the treatment group were excluded from the analysis as they had ocular conditions other than refractive error (eg, nystagmus) confirmed in their medical notes, leaving 368 children in the treatment group and 433 in the comparison group. Of 368, 230 (62.5%) of children in the treatment group had attended for the initial cycloplegic examination and been prescribed spectacles,

3/368 (0.8%) attended but no cycloplegic refraction information was available, 23/368 (6.3%) had been prescribed spectacles but had not returned for follow-up VA assessment and 112/368 (30.4%) had failed to attend any appointment following vision screening. Of the 253 children in the treatment group with cycloplegic refraction results, 157/253 (62.1%) had astigmatism (>1.00 DC) either alone (n=19) or in combination with hypermetropia (>+3.0 DS) (n=56), low hypermetropia (>+1.0 DS to +3.0 DS) (n=16) or myopia (≤−0.50 DS) (n=66). Of 253, 35 (13.8%) had hypermetropia alone, 11 (4.3%) had myopia alone and 50 (19.8%) children had low hypermetropia. Of 253, 55 (21.7%) additionally had anisometropia (≥1.0 D difference). For those children with a cycloplegic refraction result (table 1) the SER ranged from −7.875 to +7.50 D in the better eye and −8.25 to +7.50 D in the worse eye. Of the 368, 14 (3.8%) children had a constant or intermittent strabismus, 5 of whom had been prescribed occlusion therapy for amblyopia at

**Table 1** Characteristics of Born in Bradford children and mothers included in the analyses

| | Comparison group n=433 | Treatment group n=368 | P values* |
|---|---|---|---|
| Children | | | |
| Age (months) year 1 | 60 (4.2) | 60 (4.5) | 0.119 |
| Gender | | | |
| Male | 229 (51.1) | 183 (49.7) | |
| Female | 219 (48.9) | 185 (50.3) | 0.693 |
| Ethnicity | | | |
| White | 125 (28.0) | 91 (24.9) | |
| Pakistani | 262 (58.7) | 232 (63.4) | |
| Other | 59 (13.3) | 43 (11.7) | 0.403 |
| Route of birth | | | |
| Vaginal | 342 (77.0) | 291 (79.7) | |
| Caesarean | 102 (23.0) | 74 (20.3) | 0.355 |
| Gestational age at birth (weeks) | 277 (12.0) | 276 (13.0) | 0.158 |
| Birth weight (g) | 3184 (550.0) | 3128 (573.0) | 0.155 |
| VA better eye | 0.113 (0.049) | 0.271 (0.138) | <0.001 |
| VA worse eye | 0.135 (0.046) | 0.428 (0.189) | <0.001 |
| SER better eye† | – | 1.19 (0.95) | – |
| SER worse eye† | – | 1.98 (1.27) | – |
| Mother | | | |
| Age (years) | 27.3 (5.4) | 28.1 (5.7) | <0.001 |
| Mother's education | | | |
| <A -level | 227 (64.5) | 190 (69.3) | |
| A-level or above | 125 (35.5) | 84 (31.7) | 0.201 |
| In receipt of means-tested benefits (yes) | 163 (45.0) | 144 (50.1) | 0.139 |

Values are numbers (%) or mean (SD).
VAs are measured in logMAR; therefore, higher values represent poorer VA.
*Difference between comparison and treatment groups ($\chi^2$ or t-test as appropriate).
†Cycloplegic results were available for the treatment group only.
logMAR, logarithm of the minimum angle of resolution; SER, spherical equivalent refraction; VA, visual acuity.

**Table 2** Baseline characteristics of participants in the treatment group retrospectively classed as adherent and non-adherent

| | Adherent n=173 (47.0%) | Non-adherent n=195 (53.0%) | P values* |
|---|---|---|---|
| **Children** | | | |
| Age (months) year 1 | 59.4 (4.5) | 59.6 (4.5) | 0.850 |
| Gender | | | |
| Male | 81 (46.8) | 102 (52.3) | |
| Female | 92 (53.2) | 93 (47.7) | 0.293 |
| Ethnicity | | | |
| White | 48 (27.9) | 43 (22.2) | |
| Pakistani | 103 (59.9) | 129 (66.5) | |
| Other | 21 (12.2) | 22 (11.3) | 0.387 |
| Route of birth | | | |
| Vaginal | 137 (79.6) | 154 (79.8) | |
| Caesarean | 35 (20.4) | 39 (20.2) | 0.973 |
| Gestational age at birth (weeks) | 276 (13.0) | 275 (14.0) | 0.383 |
| Birth weight (g) | 3121 (569.0) | 3134 (579.0) | 0.833 |
| VA better eye† | 0.292 (0.150) | 0.256 (0.129) | 0.008 |
| VA worse eye† | 0.465 (0.197) | 0.399 (0.175) | 0.001 |
| SER better eye | 1.18 (0.86) | 1.20 (1.02) | 0.960 |
| SER worse eye | 2.02 (1.20) | 1.96 (1.33) | 0.657 |
| Language ability scores‡ | 97.8 (15.6) | 96.8 (16.4) | 0.553 |
| **Mother** | | | |
| Age (years) | 28.1 (5.8) | 28.0 (5.7) | 0.845 |
| Mother's education | | | |
| <A -level | 78 (60.9) | 112 (76.7) | |
| A-level or above | 50 (39.1) | 34 (23.3) | 0.005 |
| In receipt of means-tested benefits (yes) | 61 (45.5) | 83 (55.7) | 0.087 |

Values are numbers (%) or mean (SD).
VAs are measured in logMAR; therefore, higher values represent poorer VA.
*Difference between adherent and non-adherent treatment groups ($\chi^2$ or t-test as appropriate).
†No child was wearing spectacles at the baseline assessment.
‡Age-adjusted language ability measure for British Picture Vocabulary Scale.
logMAR, logarithm of the minimum angle of resolution; SER, spherical equivalent refraction; VA, visual acuity.

follow-up after vision screening. Those children were not excluded from the analysis as they met the initial VA referral criteria and had been prescribed spectacles.

Baseline (year 1) characteristics of the children in the comparison and treatment groups are shown in table 1. A small mean difference (−0.021 logMAR, 95% CI −0.022 to −0.020) in VA between the eyes of the comparison group was found, equating to one letter difference. This is not clinically significant but is statistically significant, therefore, VAs are presented for the better and worse eye separately. Higher levels of VA were found in both eyes of the comparison group compared with the treatment group ($\chi^2$ p<0.001) (table 1). The only demographic factor found to differ between the comparison and the treatment group was the average mother's age which was around 10 months more in the treatment group ($\chi^2$ p<0.001).

Table 2 presents the baseline (year 1) characteristics of those children in the treatment group retrospectively categorised as adherent (173/368, 47.0%) and non-adherent (195/368, 53.0%) (figure 1). In the non-adherent group, no child wore spectacles at their year 2 assessment and 39/195 (20%) wore them in year 3 only. At baseline, the group subsequently classed as adherent had a lower level of VA compared with the non-adherent group in both the better and worse eye (table 2). The only other factor that differed between the adherent and the non-adherent groups was the mother's level of education with 50/173 (39.1%) of adherent children having mothers educated to A-level or above compared with only 34/195 (23.3%) of the non-adherent group ($\chi^2$ p=0.005). BPVS did not differ between the adherent and non-adherent children (p=0.553) suggesting no difference in cognitive ability.

## Visual acuity

At baseline compared with the comparison group both the adherent (mean difference: 0.337 logMAR; 95% CI 0.304 to 0.370) and non-adherent groups (mean difference: 0.273 logMAR; 95% CI 0.241 to 0.305) had lower levels of VA in the worse eye. Table 3 and figure 2 present the VA trajectories over the course of the study. These show that after adjusting for previously described early-life and maternal variables, the VA of both eyes for all three groups; the comparison, the adherent and the non-adherent groups improve over time.

The VA of all children improved with increasing age, −0.009 log units per month (95% CI −0.011 to −0.007) (worse eye) and −0.006 log units per month (−0.008 to −0.005) (better eye) (table 3).

Over and above this improvement, the adherent group (worse eye) improved by a further −0.008 log units per month (95% CI −0.009 to −0.007). The adherent children, therefore, improved overall by −0.017 (95% CI −0.020 to −0.015) log units per month (approximately, two letters every 3 months) and also demonstrated a small amount of improvement in the better eye above that expected from age (table 3).

The non-adherent group (worse eye) improved by −0.003 log units per month (95% CI −0.004 to −0.001) above that expected from age. The non-adherent children, therefore, improved overall by −0.012 log units per month (95% CI −0.014 to −0.010). No additional improvement above that expected from age was demonstrated in the better eye (table 3).

Sensitivity analysis redefining the classification of adherence did not materially affect the results.

## Literacy

The unadjusted model shows the final letter-ID score reduces by −0.9 units (95% CI −1.15 to −0.64) for every one line (0.10 logMAR) fall in VA of the better eye (table 4). This association persists but is weaker after fully adjusting for the socioeconomic and demographic factors, the letter-ID score declines by −0.327 units (95% CI −0.540 to −0.115) for every one line fall in VA. Separate adjusted analysis of the VA level of the worse eye shows similar results but with weaker association, letter-ID score declines by −0.260 units (95% CI −0.414 to −0.105) for every one line fall in VA.

Children of mothers educated to A-level or above had increased letter-ID scores (0.765 units; 95% CI 0.156 to 1.374) compared with those with lower qualifications. Ethnicity other than white British or Pakistani heritage was associated with better letter-ID score, which might reflect the higher number of mothers educated to above A-level in this group. Greater birth weight was also associated with increased letter-ID score (table 4). Adjustment for SER made no difference and was not associated with letter-ID (p=0.306). It was therefore not included in the models. Similarly, subsequent analysis replacing SER with refractive error categories did not show an association with letter-ID (online supplementary information 2).

**Table 3** Change in visual acuity for the better and worse eye over time by group: comparison, adherent and non-adherent

| | Unadjusted (worse eye) (95% CI) | Adjusted† (worse eye) (95% CI) | Unadjusted (better eye) (95% CI) | Adjusted† (better eye) (95% CI) |
|---|---|---|---|---|
| Constant | 0.177 (0.159 to 0.194)* | 0.386 (0.124 to 0.648)* | 0.240 (0.026 to 0.454)* | 0.240 (0.026 to 0.454)* |
| Age (months) | −0.009 (−0.011 to −0.008)*** | −0.009 (−0.011 to −0.007)*** | −0.006 (−0.008 to −0.005)*** | −0.006 (−0.008 to −0.005)*** |
| Age (months) squared | 0.00016 (0.00012 to 0.00020)*** | 0.00016 (0.00012 to 0.00021)*** | 0.00010 (0.00007 to 0.00013)*** | 0.00010 (0.00006 to 0.00014)*** |
| Group (reference: comparison) | | | | |
| Adherent | 0.337 (0.309 to 0.366)*** | 0.337 (0.304 to 0.370)*** | 0.184 (0.162 to 0.106)*** | 0.170 (0.144 to 0.196)*** |
| Non-adherent | 0.277 (0.250 to 0.305)*** | 0.273 (0.241 to 0.305)*** | 0.150 (0.128 to 0.172)*** | 0.148 (0.123 to 0.174)*** |
| Age × group interaction | | | | |
| Age × adherent | −0.008 (−0.009 to −0.007)*** | −0.008 (−0.009 to −0.007)*** | −0.004 (−0.005 to −0.004)*** | −0.004 (−0.005 to −0.003)*** |
| Age × non-adherent | −0.003 (−0.003 to −0.001)*** | −0.003 (−0.004 to −0.001)*** | −0.001 (−0.002 to 0.000) | −0.002 (−0.004 to 0.000) |

× : interaction between group and age to determine if the effect of being in a particular group changes with age. The total effect for any one group is the coefficient for age plus the additional effect of age for that group.
*P<0.05, **P<0.01, ***P<0.001.
†Model adjusted for gender, ethnicity, gestation period, birth weight, birth route, maternal education status, maternal age and means-tested benefit status.

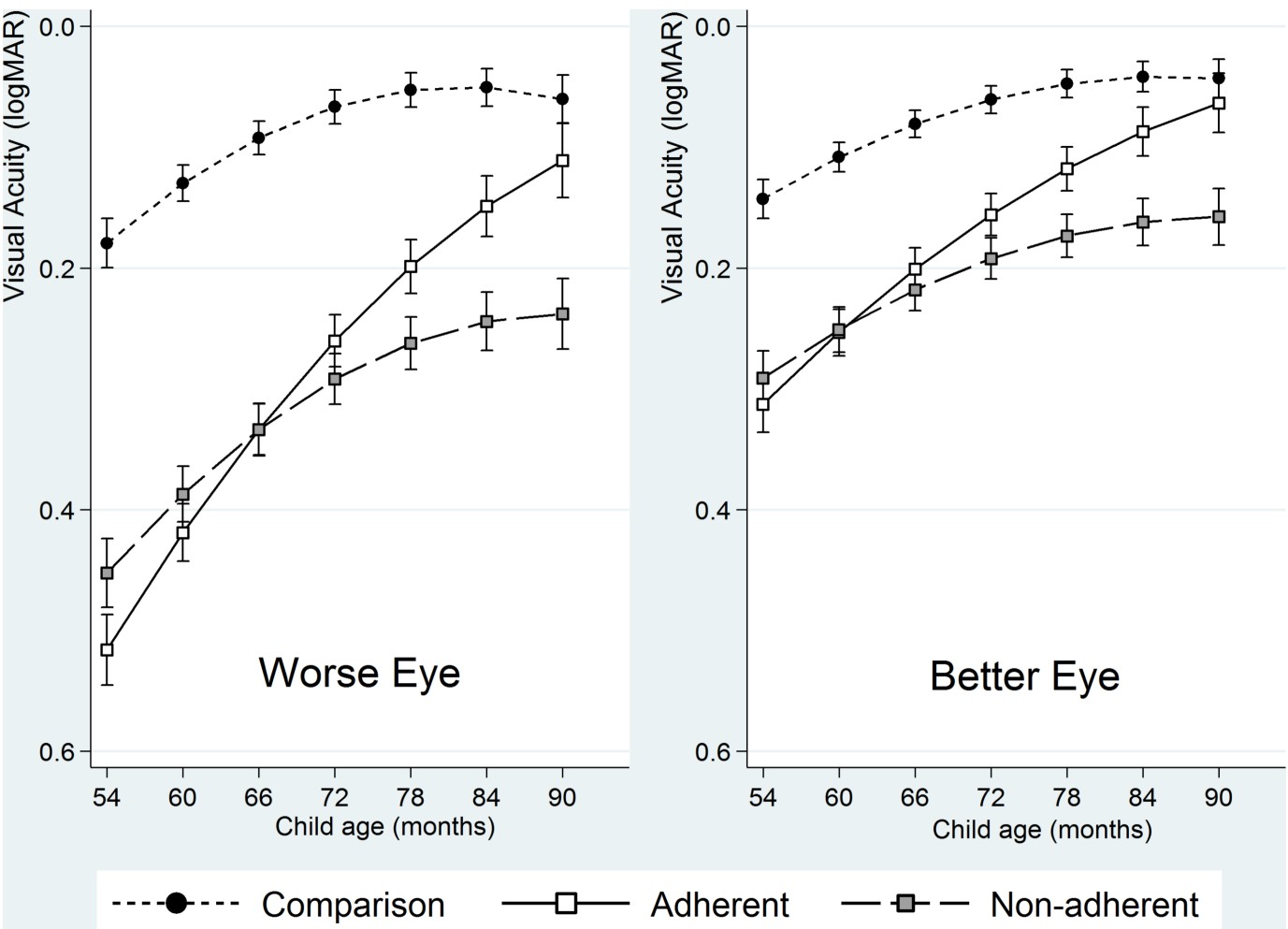

**Figure 2** Projected visual acuity (logMAR) trajectory (with 95% CIs) by group over time (child's age in months) for the better and worse eye, fully adjusted for all early-life and maternal covariates. logMAR, logarithm of the minimum angle of resolution.

A predictive model of the letter-ID score over time for children in each group (figure 3) was constructed using both the unadjusted and adjusted data from the VA trajectories (table 3) and incorporated into the model reporting letter-ID (table 4). The unadjusted trajectory shows both adherent and non-adherent groups at baseline have lower letter-ID scores than the comparison group. The predicted trajectory of improvement in the adherent group is greater than the non-adherent group with the later letter-ID scores of the adherent group converging on those of the comparison group by year 3. The non-adherent group although improving over time does not catch up with the adherent or the comparison groups. After adjusting for socioeconomic and demographic variables, the trend is similar but with a smaller difference between the groups.

Table 5 presents the effect size of wearing spectacles on the letter-ID scores between the groups annually over the 3 years of the study. Comparing the letter-ID scores between the adherent and the non-adherent group, a gradual increase in the effect size over time is demonstrated with the greatest effect size (0.11) between the adherent and non-adherent groups shown in year 3.

### VA: year 3

The results demonstrate a statistically significant correlation between near and distance VA in year 3 (right eye, r=0.663 and left eye, r=0.642) (online supplementary information 3). In addition, the associations between the near VA and literacy score and distance VA and literacy score are approximately the same (online supplementary information 4).

### DISCUSSION

This is the first longitudinal study to assess the effect of adherence/non-adherence to spectacle wear on VA and literacy in children following vision screening. The VA of children who adhered to spectacle wear was found to improve at a far greater rate compared with those who were non-adherent, with the VA of adherent children reaching similar levels to the VA of the comparison children by the end of the study. Our results further indicate that early developing literacy is affected by the level of VA even after adjusting for socioeconomic and demographic factors associated with educational attainment. The letter-ID score declines by approximately 1.5% for

**Table 4** Associations between letter-ID score, visual acuity (better eye), maternal and early-life factors

| Factor | Unadjusted model (95% CI) | P values | Fully adjusted model (95% CI) | P values |
|---|---|---|---|---|
| Constant | 18.82 (17.91 to 19.73) | <0.001 | −20.6 (−28.2 to −13.0) | <0.001 |
| Age | 1.30 (1.21 to 1.38) | <0.001 | 1.28 (1.19 to 1.37) | <0.001 |
| Age squared | −0.02 (−0.02 to −0.02) | <0.001 | −0.020 (−0.022 to −0.017) | <0.001 |
| Visual acuity: change in letter-ID | −0.90 (−1.15 to −0.64) | <0.001 | −0.327 (−0.540 to −0.115) | 0.003 |
| Per 0.1 log unit (one line) | | | | |
| Letter-ID baseline (year 1) | | | 0.348 (0.326 to 0.371) | <0.001 |
| BPVS | | | 0.019 (−0.001 to 0.039) | 0.064 |
| Ethnicity | | | | |
| Pakistani heritage | | | 0.668 (−0.016 to 1.353) | 0.056 |
| Other | | | 1.174 (1.159 to 2.189) | 0.023 |
| Gender | | | | |
| Female | | | 0.471 (−0.093 to 1.035) | 0.102 |
| Birth weight (per 100 g) | | | 0.074 (0.008 to 0.141) | 0.029 |
| Gestational age (weeks) | | | −0.053 (−0.257 to 0.151) | 0.611 |
| Receiving benefits | | | −0.086 (−0.661 to 0.4990) | 0.770 |
| Mothers level of education | (higher than A-level) | | 0.765 (0.156 to 1.374) | 0.014 |
| Mothers age at birth (years) | | | −0.048 (−0.100 to 0.005) | 0.075 |

BPVS, British Picture Vocabulary Scale (baseline standardised score); ID, identification.

every one line of reduction in VA. In this and similar populations,[14 31] where children have been reported to have reduced VA levels (>0.30 logMAR in better eye), there is likely to be an impact on developing literacy skills. The effect size (0.11) of being adherent to spectacle wear compared with non-adherence in year 3 of our study is the same as that reported in a Chinese study providing free spectacles to children[32] and is comparable with reported educational interventions.[33] Thus, children who fail vision screening and adhere to spectacle wear have the potential to improve their VA, further influencing early literacy development.

Adherence to spectacle wear is highly influenced by socioeconomic and demographic factors, particularly maternal education, a factor that is also known to be associated with educational attainment.[34] Children with reduced VA and who are in less educated families are less likely to adhere to treatment, which will further impact on their educational attainment and future life chances. We were, however, able to adjust for the many associated maternal and early-years factors, the value of embedding this study within a birth cohort. A study examining academic performance in US schools reports that failing vision screening was predictive of being in the lowest quartile of academic performance.[35] Conversely, a longitudinal study of children aged 9–10 years in Singapore, Dirani *et al*[36] found VA did not play a significant role in predicting academic performance. However, the children were older, mainly myopic and only a small number of participants had decreased VA which may account for the difference in their findings relative to ours.

The VA of children in all groups (adherent, non-adherent and comparison group) continued to improve throughout this study. The improvement in VA found in the comparison group is similar to that reported for normal visual development, with optimum VA achieved around 6 years of age.[37 38] The improvement in VA of the worse eye found in adherent children over the time of the study was significantly greater than that expected solely from visual development[39] or indeed from retest variability[40] and was almost double that of the comparison group. Little additional improvement above that expected from visual development was demonstrated in the worse eye of the non-adherent children, an indication that the improvement in the adherent children is not due to regression to the mean. The longitudinal observation of the children demonstrates improvement in VA and in literacy, with the non-adherent group demonstrating persistently lower literacy scores throughout the study, although the effect is attenuated after adjusting for other factors. Annual improvement in academic achievement is well recognised and is particularly notable in the early years of schooling with the initial improvement thought to be associated with the effect of entering school, combined with rapid early child development followed by a plateau in academic growth as children progress through school grades.[20]

Early literacy development is complex and associated with socioeconomic and demographic factors, in particular maternal education. However, even after taking these factors into account VA continues to be associated with literacy; the poorer the level of VA, the greater the

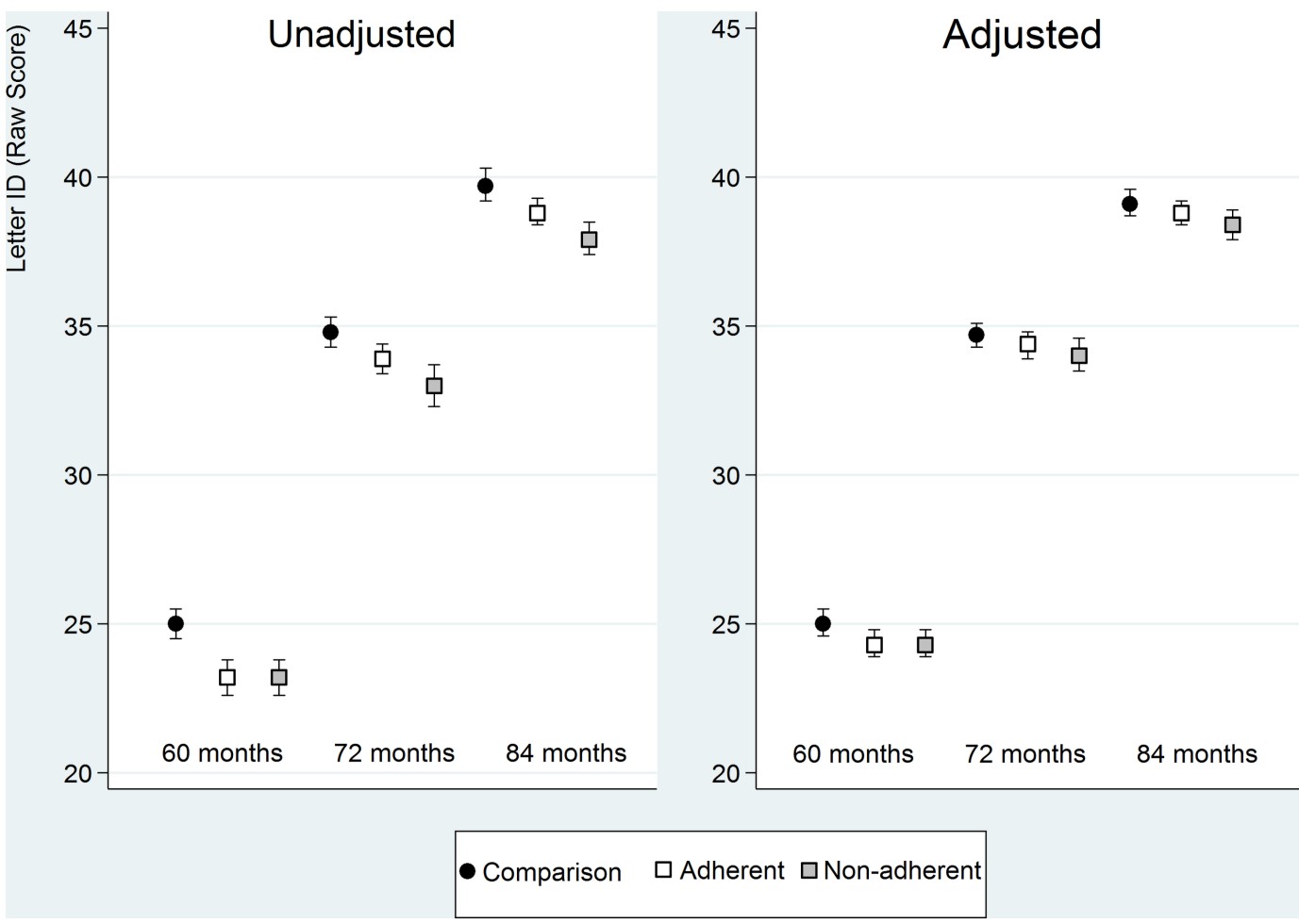

**Figure 3** Predicted letter-ID scores over time (child's age in months) based on the trajectories of the visual acuity (adjusted model) of the better eye. The adjusted model includes all early-life and maternal covariates for the comparison, adherent and non-adherent groups. ID, identification.

reduction in the literacy score. In a Singaporean study,[39] a strong association between paternal level of education and academic school performance was reported. As one might expect, higher levels of maternal education have a positive impact on literacy.[41 42] In addition, mothers with higher educational attainment are more likely to effectively access health services, and are more likely to adhere to prescribed treatment.[43]

**Table 5** Annual Literacy Scores by group

| Year | Group | Letter-ID (raw score) | Comparison groups | Effect size (Cohen's d)* |
|------|-------|----------------------|-------------------|--------------------------|
| 1 | Comparison | 25 | Comparison versus adherent | 0.06 |
| | Adherent | 24.3 | Comparison versus non-adherent | 0.06 |
| | Non-adherent | 24.3 | Adherent versus non-Adherent | 0.00† |
| 2 | Comparison | 34.7 | Comparison versus adherent | 0.05 |
| | Adherent | 34.4 | Comparison versus non-adherent | 0.13 |
| | Non-adherent | 34.0 | Adherent versus non-Adherent | 0.07 |
| 3 | Comparison | 39.1 | Comparison versus adherent | 0.08 |
| | Adherent | 38.8 | Comparison versus non-adherent | 0.18 |
| | Non-adherent | 38.4 | Adherent versus non-Adherent | 0.11 |

SD 10.9 at year 1, 5.6 at year 2 and 3.8 at year 3.
*Based on group difference divided by the pooled SD of letter-ID score.
†In year 1, there is no difference as spectacle wear has not commenced.
ID, identification.

Our study shows an association between VA and literacy score but no association between SER and literacy. Neither did further analysis by refractive error types indicate an association with literacy, this is most likely related to a lack of power due to the small numbers when refractive error is categorised in our study. Our findings differ from previous studies reporting an association between refractive error and literacy.[11 12]

Hypermetropia has been reported to be associated with poor literacy. A large cross-sectional American study (vision in preschoolers - hyperopia in preschoolers (VIP-HIP)) of preschool children aged 4–5 years found that children with uncorrected hypermetropia in conjunction with reduced binocular near VA (worse than 20/40) have poorer literacy than those with hypermetropia and a good level of binocular near VA.[12] The VIP-HIP study reports that the level of binocular near VA was predictive of literacy scores; with hypermetropic children with binocular near VA better than 20/40, demonstrating literacy scores similar to those children who were emmetropic. Although the VIP-HIP study does not report distance VA levels of the children, it does state that the analysis of the distance VA resulted in similar findings, an indication that distance VA levels may also influence early literacy scores.

Astigmatism has also been reported to be associated with reduced literacy. In native American children bilateral uncorrected astigmatism (≥1.00 DC) has been reported to reduce reading fluency, and children with moderate astigmatism are reported to have lower VA and fluency than those with no or low astigmatism.[11] The findings reported from both the above studies may indicate that moderate to high degrees of uncorrected hypermetropia or astigmatism which reduce VA is associated with a reduction in literacy scores.

Classroom-based tasks where fixation frequently changes are reported to require high levels of distance VA (0.33 logMAR) and slightly lesser levels of near VA (0.72 logMAR),[44] this is most probably due to print size for early readers being enlarged. We would suggest therefore that where VA is reduced beyond that required in the learning environment, it will impact on a child's developing literacy and hence the association we report between distance VA and literacy.

The longitudinal design of this study provides an insight into development of VA and literacy in the early years of schooling, and the use of linked data from the mothers and children participating in the BiB cohort study permitted the many potential confounding factors associated with educational attainment to be accounted for. We include children with a wide range of refractive error and VAs allowing a robust analysis of the influence of both factors on developing literacy. The study does however have some weaknesses. It is not a randomised controlled trial and non-adherence was defined retrospectively by the failure of the child to wear their prescribed glasses at one assessment; it is possible that this was a unique event and is not representative of the child's true adherence to spectacle wear over the course of the study. If this is

indeed the case, then the random misclassification is likely to underestimate the difference found between the adherent and non-adherent groups.[45] In addition, the sensitivity analysis redefining non-adherence does not demonstrate any material difference in the results.

A cycloplegic examination was not undertaken for all children and there will be some children with reduced vision who were not identified at screening (false negatives). No child who had a cycloplegic refraction was found to be a false positive but a proportion of the children who failed to attend for the cycloplegic examination may be false positives. This misclassification will similarly be random, underestimating the size of estimates of effect and suggests our estimates may be conservative.[45]

VA is the sole measure of visual function reported from the study and it is possible other measures of visual function are also associated with academic performance; further research would be required to explore these associations. The VA assessment and the literacy test are both letter based and children who struggle with letter-ID may also demonstrate a poor ability with the VA test. However, all children used a matching technique, a skill that is present in children as young as 3 years[46] and no child who failed the screening was classed as false positive.

During visual maturation, the presence of neurodevelopmental disorders such as refractive error, and strabismus may contribute to a reduction in VA and early intervention is required. This study demonstrates that wearing spectacles is an effective intervention to improve VA, and that this will impact positively on developing literacy. The children who do not adhere to spectacle wear are likely to be those in families who are less well educated. Further research is required to better understand the reasons for non-adherence and evaluate interventions to promote adherence to spectacle wear. This has the potential to improve vision and support future life chances in children who may already face educational disadvantage.

**Acknowledgements** We thank all the families and schools who took part in this study, the orthoptists from Bradford Teaching Hospitals Foundation Trust who conducted the vision screening programme, the researchers from the Starting Schools programme who collected the literacy measures, Patrick Friis, Alexandra Morris and Hannah Farrugia who collected follow-up measures and the Data Support Team from Bradford Institute for Health Research who created and maintain the data linkage system.

**Contributors** AB initiated the project, designed data collection, monitored data collection for the whole study, wrote the statistical analysis plan, cleaned and analysed the data, and drafted and revised the paper. She is the guarantor. BK wrote the statistical analysis plan, cleaned the data and revised the draft paper. BC initiated the project and revised the draft paper. BTB contributed to the design of the study and revised the draft paper. MB contributed to the design of the study and revised the draft paper. JB contributed to the design of the study and revised the draft paper. TAS initiated the project, wrote the statistical analysis plan and revised the draft paper.

**Funding** AB is funded by a National Institute for Health Research Post-Doctoral Fellowship Award (PDF-2013-06-050). The Born in Bradford study presents independent research commissioned by the National Institute for Health Research Collaboration for Applied Health Research and Care (NIHR CLAHRC) and the Programme Grants for Applied Research funding scheme (RP-PG-0407-10044).

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
