## [Reviewer comments · BMJ Open]

ARTICLE DETAILS

TITLE (PROVISIONAL)	Effect of adherence to spectacle wear on early developing literacy: A longitudinal study based in a large multi-ethnic city, Bradford, UK.
AUTHORS	Bruce, Alison; Kelly, Brian; Chambers, Bette; Barrett, Brendan; Bloj, Marina; Bradbury, John; Sheldon, Trevor

VERSION 1 – REVIEW

REVIEWER	Shelley Hopkins QUT, Australia
REVIEW RETURNED	31-Jan-2018

GENERAL COMMENTS	Reviewer's report: Effect of adherence to spectacle wear on early developing literacy: a longitudinal study Overall, this paper addresses a gap in the literature relating to adherence to spectacle wear – and effect on visual acuity, as well as early literacy development. The paper is very well-written and enjoyable to read/review. A strength of the study is the cohort, i.e. the ability to control for many epidemiological confounders, given it is part of the BiB study. Line by line comments follow: Introduction 1. The opening paragraph has a strong amblyopia focus, whereas the study itself doesn't identify amblyopia (rather strabismus and reduced va – which may correct with specs); could the authors in this opening paragraph speak more broadly about the goals of vision screenings, i.e. identify children with reduced va, as well as amblyogenic risk factors.2. The end of the 3rd sentence doesn't read well. It is unclear whether detect and treat early relates to amblyopia only, or refractive error more broadly3. The opening sentence in para 2 states that decreased VA is associated with reduced literacy levels. Is this distance or near va? Conceptually, reduced near va would seem to be more significant with literacy levels, compared with distance va; this is supported by the VIP-HIP study, as well as studies that have not found reduced literacy in children with distance va – e.g. uncorrected myopes, where near va
--

remains unaffected. Maybe some preliminary discussion on the limitation of current screenings, that measure distance va alone (and don't consider near va) could be included here

Methods

1. Is it correct that the comparison group did not undergo cycloplegic refraction? This would seem a significant limitation of a longitudinal study. The following factors need considering/discussing: at time point 3, there may be children in the comparison group that have developed myopia – and had reductions in va – how will this affect the interpretation; also, by not cyclopleging the comparison group a proportion of low-mod hyperopes may be missed (which links to the limitation of standard vision screenings – not considering the impact on near visual function of hyperopia) – how will this affect the interpretation – specifically with regards to literacy;
2. Were other tests of near visual function included in the baseline assessment – e.g. accommodative response, amps, etc., this may potentially provide more information on the link between the SER and VA findings (and the interesting differences found between the adherent and non-adherent group) as well as explain why there was no link between SER and literacy – which is unusual given the large body of evidence linking hyperopia with literacy; subgroup analysis of SER and near function, or SER and NVA may strengthen some of the findings reported in this study
3. It would be useful to have more information included on the prescribing philosophy in the treatment group; what cut-offs were used, what did the 'clinical judgment' consider specifically, what adjustment/if any was made in the final prescription following cycloplegic refraction, were all the low hyperopes (+1.00 - <+3.00) prescribed glasses, etc...
4. In the Follow-up Assessments (Yrs 2 and 3), a Bailey-Lovie near-vision chart is introduced – this is the first time that NVA is mentioned in the study. Was it also measured at baseline?
5. At the Yr 2 and 3 follow-up, were cycloplegic refractions performed? If not, how have the authors determined that there has been no change in rx resulting in reduced visual acuities? Similarly, did any of the participants bring in different glasses at the Year 2 and 3 follow ups, from outside optometrists? Were the power of any different glasses measured – to assess whether there had been a change in SER from baseline?
6. Statistical analysis – I agree that the classifying of adherent and non-adherent is a limitation of the study, but I think that it may be underestimating the effect of both the non-adherence and adherence groups? Children may be classified in the adherence group who by chance had spectacles at the two times of testing, but are otherwise non-adherent, which would bring down the effect of the adherent group as well. Was a separate analysis performed where non-adherence was classified as 0/2 times wearing

specs, and adherence 2/2?

Results

1. Of the 368 children in the treatment group, it reads that all children were prescribed spectacles? Were there any children that were false positives from the initial screenings – and any emmetropes, i.e. less than 1.00D?
2. On page 12, it is interesting that 20% of children were still wearing glasses in Year 3 but were classified in the non-adherent group (based on not having glasses at the Year 2 follow-up). In my clinical experience, it would seem that the child must be somewhat adherent to spectacle wear if they are still presenting with spectacles two years following the initial examination (or they have been seen elsewhere in the meantime). This is the group that may benefit from re-analysis, so just the 0/2 and 2/2 compliance groups are included.
3. Table 2 – it is an interesting finding that the VAs for both better and worse eyes are different between in the adherent and non-adherent groups, yet there is no difference in SER between groups. Can the authors provide any comment as to what they think might be driving this?
4. The results around VA improvement between comparison and treatment groups, and worse and better eyes are a little confusing. They are presented 3 times (abstract, in results with Table 3 and discussion). After reading the abstract, my initial interpretation was that the VA in the better eye had biggest gains in the adherence group, compared with the worse eye, as well as with the non-adherent and comparison group. Then reading the results section, my interpretation changed to the worse eye having the most gains. The summary of this section in the discussion, was the easiest to understand. If the authors could potentially find a more simple way of presenting this group of results it would make the interpretation a lot easier.
5. In the literacy section, there was no adjustment for SER as it was not associated with letter-ID. Was any separate analysis completed looking specifically at hyperopia and letter-ID, or hyperopia and reduced NVA and letter-ID (VIP-HIP). One might expect differences in association between SER and letter-ID in the hyperope/reduced NVA group, compared with a myopic/reduced DVA group (as per the Dirani paper cited later in the discussion)

Discussion

1. The opening sentence states that this study is the first to assess VA and literacy in children following screening. There are lots of other studies that have previously investigated this (as the authors discuss later in the discussion). The uniqueness of this study is its attempt to measure adherence/non-adherence to spectacle wear and effect on VA and literacy.
2. The authors cover a large number of studies in the discussion. Description of a lot of the studies are minimal, leaving the reader questioning how the findings directly

	relate to the current study. Perhaps a few of these studies could be removed, and some of the larger studies discussed in more detail. E.g. the large VIP-HIP study currently has a similar amount of explanation as a small pilot study.  3. Reference number 34, that uses the addition of +/- 2.00D lenses – it is not clear to me what this means, so needs more explanation or removing. 4. The sentence on reference number 45 also is not clear, so needs more explanation or removing. 5. In the 3rd last paragraph, the term moderate hypermetropia is used to describe the range of refractive error in the current study. This is the first time 'moderate' has been used to classify the RE range; in the results low hypermetropia and hypermetropia (>+3.00 are used) 6. Given the large body of evidence linking uncorrected hypermetropia and reading performance, I think the lack of association between SER and literacy in the current study needs covering more in the discussion 7. In the second last paragraph, it says DVA is the sole measure of visual function, but NVA is included in the methods sections at follow-up 2 and 3.
--	---

REVIEWER	Geoff Bradford, MS, MD West Virginia University, USA
REVIEW RETURNED	07-Feb-2018

GENERAL COMMENTS	General comments: This is a nice prospective longitudinal study over 2 years examining the association between visual acuity, glasses wear and literacy development in a multi-ethnic cohort of 433 children 4-5 yrs of age who failed a vision screening and were compared to 368 randomly selected children of the same ages who passed the screening. The study compared the results of a visual acuity test with a letter identification test of early literacy. Treatment subjects (those who failed screening) were divided into two groups: those adherent to wearing glasses and those not adherent to glasses. The literacy performance of these two groups was assessed relative to a no-spectacle-wear comparison group. The study concluded that early literacy is associated with the level of visual acuity and that children who adhered to wearing glasses improved both their vision and their potential to improve literacy. Specific comments: Page 7: Line 9. Visual acuity screening in preschool children is known to produce significant false negative and false positive results. How does one know if the children who passed the vision screening did not, in fact have a significant refractive error? How many children who failed the screening did not have a significant refractive error? Line 29. Can you include a figure of the LogMAR eye chart used in this study so readers internationally know exactly how it is displayed and formatted? Line 39. Which model of welch-Allyn auto refractor was used? Was
---

	this the Welch-Allyn “Spot” device? Line 45. Spell out NSC. Can you show these criteria in a table? Line 53. How many ophthalmologists and optometrists participated in this study? Was there a significant difference in the spectacle prescribing habits of optometrists vs ophthalmologists? Page 9: Line 15. Can you include a figure of the Bailey-Lovie near vision chart, again for international readers who may be unfamiliar with it? Line 41: I am uncomfortable with the definition here of adherence being spectacle wear at the time of follow up assessment. What about all other days up to this assessment? Page 11: Line 13. Delete the word for “attended the initial cycloplegic exam. . . “ Lines 15-19: I would also exclude from this study those children who had incomplete assessments; ie 1) no cycloplegic refraction, 2) No follow up visual acuity, 3) failed to attend follow up appts. Or justify their inclusion. Line 27: The definition for myopia starting at only 0.5 DS sphere is rather mild, especially if one is looking at literacy related to refractive error. Children who have only 0.5 DS will invariably still have normal vision for reading and other near work, even if they do not have glasses. Line 37. Occlusion therapy is usually prescribed for amblyopia, not strabismus. If the 14 children with strabismus, also had amblyopia, this should be stated. Page 18: Line 33. Change the word reduces to declines (... by approximately 1.5%. . .) Page 19: Line 17. The +2.00D lens test has never been validated as a useful or effective test for hyperopia. Page 21: Line 13. I agree that near acuity testing, as opposed to or in addition to distance acuity testing, would have been a very nice (and perhaps very important) variable to add to this study.
--	---

VERSION 1 – AUTHOR RESPONSE

We thank the reviewers for their detailed and constructive criticism of our manuscript. Firstly we have responded to the points raised by both reviewers, we then respond on a point by point basis to each reviewer individually. We have revised the manuscript in light of the points raised and believe that the revised manuscript is now clearer as a result.

General Point 1.

Both reviewers highlighted that the definition of adherence and non-adherence may affect the result. We have therefore performed a sensitivity analysis and analysed the data using three definitions of adherence and non-adherence:

Original definition (A): Analyses and results as in Table 3 of the original manuscript (copied below).

Revised Definition (B): Adherent group = children who were adherent at both follow-up time points. Non-adherent group = non-adherent at both follow-up time points.

(This analysis **excludes** the small number (n=39) of non-adherent children at follow up 1 and adherent at follow up 2 from the non-adherent group.)

The results are very similar to analyses A, with a very slight reduction in the size of the effect for non-adherence. (Analysis B, see below).

Revised Definition (C): Adherent group = children who were adherent at one or both time points. Non-adherent = non-adherent at both time points. (No exclusions).

The results are very similar to analysis A, but this time with a very slight reduction in the size of the effect for adherence. (Analysis C, see below).

In general the differences arising from different grouping criteria are very small; and do not change the substantive interpretation derived from the initial categorisation presented in the original paper. We have included the different analyses below for the reviewers and state in the revised manuscript that a sensitivity analysis was performed (revised manuscript page 10) and that the redefining of adherence classification did not materially affect the results (revised manuscript page 15).

Analysis A: As reported in the original paper

GROUP1	WORST EYE				BEST EYE				GRO UP	Glasses at Follow Up	
	b	min9 5	max9 5	p	b	min9 5	max9 5	p		Ye ar 2	Ye ar 3
Constant	0.386	0.124	0.648	0.004	0.240	0.026	0.454	0.028	Adher e		
Child age (months)	-0.009	-0.011	-0.007	<0.001	0.006	0.008	0.005	<0.001	Non adher e		
Child age (months) squared	0.00016	0.00012	0.00021	<0.001	0.0001	0.00006	0.00014	<0.001	Non adher e		
Group (ref: comparison)									Non adher e		
Treatment - adherent	0.337	0.304	0.370	<0.001	0.170	0.144	0.196	<0.001	Adher e		
Treatment - non adherent	0.273	0.241	0.305	<0.001	0.148	0.123	0.174	<0.001	Non adher e		
Interaction: Child age*											

Group								
Age*Treatment	-	-	-	<0.001	0.004	-	-	<0.001
adherent	0.008	0.009	0.007			0.005	0.003	
Age*Treatment non	-	-	-	<0.001	0.001	-	-	0.061
adherent	0.003	0.004	0.001			0.002	0.000	

Group
 * Adherence = 2/2 times wearing glasses
 * Non Adherence = not wearing glasses at both

Analysis B: With non-adherence more tightly defined (non-adherent on both follow up occasions)

GROUP2	WORST EYE				BEST EYE				GROUP2	Glasses at Follow Up	
	B	min95	max95	p	b	min95	max95	p		Year 2	Year 3
Constant	0.316	0.066	0.565	0.013	0.186	0.019	0.391	0.075	Adhere		
Child age (months)	-0.009	-0.011	-0.008	<0.001	0.006	-0.008	-0.005	<0.001	Non adhere		
Child age (months) squared	0.00017	0.00012	0.00022	<0.001	0.0001	0.00006	0.00014	<0.001			
Group (ref: comparison)											
Treatment - adherent	0.336	0.304	0.367	<0.001	0.170	0.145	0.195	<0.001			
Treatment - non adherent	0.242	0.209	0.276	<0.001	0.131	0.104	0.157	<0.001			
Interaction: Child age*Group											
Age*Treatment adherent	-0.008	-0.009	-0.007	<0.001	0.004	-0.005	-0.003	<0.001			
Age*Treatment non adherent	-0.002	-0.003	-0.001	0.006	0.000	-0.001	-0.001	0.470			

Group 2

* Adherence = 2/2 times wearing glasses (AS BEFORE)

* Non Adherence = 0/2 times not wearing glasses at both follow up occasions (recode those who were wearing in Year2 as missing) - i.e. EXCLUDE

Analysis C. With adherence more loosely defined (adherent on at least one follow up occasion)

GROUP3	WORST EYE				BEST EYE				GROUP3	Glasses at Follow Up	
	b	min95	max95	p	b	min95	max95	p		Year 2	Year 3
Constant	0.344	0.078	0.611	0.011	0.217	0.003	0.431	0.047	Adhere	Adhere	
Child age (months)	-0.009	-0.011	-0.007	<0.001	0.006	-0.008	-0.005	<0.001	Adhere	Adhere	
Child age (months) squared	0.00017	0.00012	0.00021	<0.001	0.0001	0.00006	0.00014	<0.001	Adhere	Adhere	
Group (ref: comparison)									Non adhere	Non adhere	
Treatment - adherent	0.316	0.284	0.349	<0.001	0.164	0.138	0.190	<0.001	Adhere	Adhere	
Treatment - non adherent	0.273	0.240	0.305	<0.001	0.149	0.123	0.175	<0.001	Adhere	Adhere	
Interaction: Child age* Group											
Age*Treatment adherent	-0.007	-0.009	-0.006	<0.001	0.004	0.005	0.003	<0.001	Adhere	Adhere	
Age*Treatment non adherent	-0.003	-0.004	-0.001	<0.001	0.001	0.002	0.000	0.065	Adhere	Adhere	

Group 2

* Adherence = Wearing glasses at least once in follow up

* Non Adherence = 0/2 times not wearing glasses (recode those who were wearing in Year2 as missing) - i.e. to Adhere

General Point 2.

The measurement of near visual acuity was raised by both reviewers, although primary vision screening in the UK does not include a test of visual acuity at near we did collect near visual acuity at time point 3 only. As a result we are unable to provide a longitudinal analysis. We do however have both near VA and distance VA measures at time point 3, we therefore examined the correlation between the near and distance VA at time point 3 and also examined the association between near VA and literacy to see if the results differed from the association between distance VA and literacy at time point 3.

The results demonstrate a strong and statistically significant correlation¹ between near and distance visual acuity at time point 3 (Right Eye, $r = 0.663$; Left Eye, $r = 0.642$). In addition the associations between the near VA and literacy score and distance VA and literacy score are very similar. The results of the cross-sectional analysis (time point three) are now included as supplementary material and the results of the correlation are stated in the results section of the revised manuscript (page 18) and discussion regarding near and distance VA is now included (page 21 and 22).

Reference:

1. Evans, J. D. (1996). Straightforward statistics for the behavioral sciences. Pacific Grove, CA: Brooks/Cole Publishing.

Association of distance and near visual acuity (better and worse eyes) with Letter ID score at time point three (T3).

	Correlation with Letter ID standardised score at T3	
	r	p-value
Visual Acuity (distance) - Better eye	-0.145	< 0.001
Visual Acuity (distance) - Worse eye	-0.183	< 0.001
Visual Acuity (near) – Better eye	-0.115	0.006
Visual Acuity (near) - Worse eye	-0.140	< 0.001

General Point 3.

Spherical equivalent refraction (SER) is used in our original analyses but does not demonstrate an association with the letter ID score and was therefore dropped from the multi-variable model. Both reviewers suggested that it would be useful to examine refractive error by categories. We have therefore performed additional analyses categorising refractive error; hypermetropia only ($>+3.0$ DS, excludes astigmatism), myopia only (≤ -0.50 DS, excludes astigmatism), low hypermetropia only ($>+1.0$ to $+3.0$ DS, excludes astigmatism) and astigmatism (>1.0 DC).

We re-estimated the statistical model (original manuscript Table 4) for each type of refractive error category. The results of the sub-group analyses are shown below. The results do not demonstrate

any significant association between the individual refractive categories and the Letter ID score. This is most likely to be related to a lack of power due to the small numbers when refractive error is categorised. The additional analysis shown below for the reviewers is now included as supplementary material in the results section (page 18) of the revised manuscript and included in the discussion (page 21).

Associations between Letter-ID score and refractive error types.

FACTOR	FULLY ADJUSTED MODEL (95% CI)	p value
Constant	-21.4 (-29.0 to -13.8)	<0.001
Age	1.32 (1.23 to 1.41)	<0.001
Age squared	-0.021 (-0.023 to -0.018)	<0.001
Astigmatism	-0.329 (-0.933 to 0.275)	0.286
Hypermetropia	-1.071 (-2.586 to 0.444)	0.166
Myopia	1.386 (-2.953 to 5.275)	0.531
Low hypermetropia	0.255 (-0.835 to 1.344)	0.647
Letter ID baseline (Year 1)	0.346 (0.323 to 0.369)	<0.001
BPVS	0.024 (0.004 to 0.044)	0.019
Ethnicity		
Pakistani heritage	0.569 (-0.128 to 1.267)	0.11
Other	1.057 (0.037 to 2.078)	0.042
Gender		
Female	0.667 (0.102 to 1.232)	0.021
Birth weight (per 100g)	0.074 (0.007 to 0.14)	0.029
Gestational age (weeks)	-0.04 (-0.244 to 0.163)	0.698
Receiving Benefits	-0.011 (-0.588 to 0.565)	0.969
Mothers Level of Education (higher than A-level)	0.717 (0.11 to 1.325)	0.021
Mothers age at birth (years)	-0.054 (-0.107 to -0.002)	0.042

General Point 4.

Vision screening in young children is known to produce false negative and false positive results. Both reviewers questioned how the study identified these children and the effect on the results.

We have evaluated the Bradford population-based screening service (this paper is currently under review) which includes all the children in the city, and presents the prevalence of children failing vision screening and the risk factors for failing vision screening. The screening programme was found to have a false positive rate of 7%. In the study we present here (a subset of children from the Born in Bradford cohort) none of the children were false positives (all had a reduced visual acuity on follow-up testing) and no child was classed as emmetropic. 50 children were prescribed low hypermetropic prescriptions (>+1.00 and +3.00DS).

We agree that there will be some children with reduced vision who were not identified at screening (false negative). This misclassification will be non-differential (or random)² i.e. the likelihood of being misclassified is not associated with the literacy outcome being measured. Random misclassification results in underestimating the size of estimates of effect and so if

there is any misclassification it means that our estimates are conservative. We have added this point into the strengths and weakness section of the discussion (page 22).

Reference:

2. Flegal KM, Brownie C, Haas JD. The effects of exposure misclassification on estimates of relative risk. *Am J Epidemiol* 1986;123:736–51.

Response to Reviewer(s)' Comments:

Reviewer 1:

Introduction

1. The opening paragraph has a strong amblyopia focus, whereas the study itself doesn't identify amblyopia (rather strabismus and reduced VA – which may correct with specs); could the authors in this opening paragraph speak more broadly about the goals of vision screenings, i.e. identify children with reduced VA, as well as amblyogenic risk factors.

We thank the reviewer for highlighting this, the text (page 5) in the opening paragraph has been revised to reduce the focus on amblyopia and emphasise the aim of the vision screening programme.

2. The end of the 3rd sentence doesn't read well. It is unclear whether detect and treat early relates to amblyopia only, or refractive error more broadly.

The sentence has now been revised (page 5) and this section removed from the text.

3. The opening sentence in para 2 states that decreased VA is associated with reduced literacy levels. Is this distance or near va? Conceptually, reduced near va would seem to be more significant with literacy levels, compared with distance va; this is supported by the VIP-HIP study, as well as studies that have not found reduced literacy in children with distance va – e.g. uncorrected myopes, where near va remains unaffected. Maybe some preliminary discussion on the limitation of current screenings, that measure distance va alone (and don't consider near va) could be included here.

The opening sentence in paragraph 2 has been revised (page 5). We have also taken on the recommendations of the reviewer (general point 2) and now present the results of the near VA and distance VA at time point 3 as a supplementary analysis. We further discuss the association of distance VA with literacy found in our study and compare the results to the VIP-HIP study in the discussion section of the revised manuscript (page 21).

Methods

1. Is it correct that the comparison group did not undergo cycloplegic refraction? This would seem a significant limitation of a longitudinal study. The following factors need considering/discussing: at time point 3, there may be children in the comparison group that have developed myopia – and had reductions in va – how will this affect the interpretation; also, by not cyclopleging the comparison group a proportion of low-mod hyperopes may be missed (which links to the limitation of standard vision screenings – not considering the impact on near visual function of hyperopia) – how will this affect the interpretation – specifically with regards to literacy.

This study observes the children as they follow the standard clinical pathway following vision screening so we were not able to perform cycloplegic refraction on children who passed the vision screening, nor did we perform a cycloplegic refraction on those children that failed to attend for any

follow-up. We agree that there may be some children who were not identified at screening with a low degree of hypermetropia. This misclassification will however be random misclassification and therefore the results we present are likely to underestimate the effect (see general point 4 above). A small number (6/433(1.4%)) of children in the comparison group were myopic at time point 3, including them in the analyses does not materially affect the results. We have performed additional analyses (General points 3 and 4) to examine the effect of the different refractive categories and now include discussion regarding refractive categories and near VA in the discussion section of the revised manuscript (page 22).

2. Were other tests of near visual function included in the baseline assessment – e. g. accommodative response, amps, etc., this may potentially provide more information on the link between the SER and VA findings (and the interesting differences found between the adherent and non-adherent group) as well as explain why there was no link between SER and literacy – which is unusual given the large body of evidence linking hyperopia with literacy; subgroup analysis of SER and near function, or SER and NVA may strengthen some of the findings reported in this study.

Clinical data from the population-based vision screening programme formed the baseline assessment; this did not include measurement of accommodation as it is not part of the screening protocol. We have responded to the reviewers' comments regarding the omission of near VA (General Point 2) and have revised the manuscript and it now contains further discussion regarding SER and VA, both near and distance (page 21 and 22).

3. It would be useful to have more information included on the prescribing philosophy in the treatment group; what cut-offs were used, what did the 'clinical judgment' consider specifically, what adjustment/if any was made in the final prescription following cycloplegic refraction, were all the low hyperopes (+1.00 - <+3.00) prescribed glasses, etc...

This study followed children through the clinical pathway following vision screening and we collected data confirming refractive prescriptions from the medical notes. The results therefore reflect the individual prescribers' clinical practice. We did not request the ophthalmic professionals to change practice, and we report their recorded results.

Low hypermetropic corrections were prescribed for 50/368 (13.3%) of children in the treatment group. The text has been revised to include "Spectacles were prescribed based on the result of the cycloplegic refraction and clinical judgement; children were generally prescribed spectacles, including low degrees of hypermetropia if they had a reduced visual acuity." (pages 7 and 8)

4. In the Follow-up Assessments (Yrs 2 and 3), a Bailey-Lovie near-vision chart is introduced – this is the first time that NVA is mentioned in the study. Was it also measured at baseline?

Primary vision screening in the UK does not include a test of visual acuity at near, we did however, collect near visual acuity at time point 3 only. In view of comments from both reviewers we have analysed the near VA data collected at time point 3 to examine the correlation between the near VA and distance VA measures (General Point 2). The results are reported (page 18) in the supplementary files and the discussion section of the revised manuscript has been amended to incorporate further discussion of near and distance VA requirements in children and their association with literacy scores (pages 21 and 22).

5. At the Yr 2 and 3 follow-up, were cycloplegic refractions performed? If not, how have the authors determined that there has been no change in rx resulting in reduced visual acuities? Similarly, did any of the participants bring in different glasses at the Year 2 and 3 follow ups, from outside optometrists? Were the power of any different glasses measured – to assess whether there had been a change in SER from baseline?

Cycloplegic refraction was not performed at time points two or three. Visual acuities at time points 2 and 3 were repeated with a pinhole if reduced (with or without spectacles, depending on whether spectacles were worn by the child at that visit) to determine if refractive correction would improve the VA. We have presented only the VA measure without the pinhole as this reflects the child's actual

level of VA whilst in school rather than their potential VA should they obtain spectacles or update spectacles.

6. Statistical analysis – I agree that the classifying of adherent and non-adherent is a limitation of the study, but I think that it may be underestimating the effect of both the non-adherence and adherence groups? Children may be classified in the adherence group who by chance had spectacles at the two times of testing, but are otherwise non-adherent, which would bring down the effect of the adherent group as well. Was a separate analysis performed where non-adherence was classified as 0/2 times wearing specs, and adherence 2/2?

In response to this comment and also similar comments from reviewer 2 regarding the definition of adherence, we have performed further analyses re-categorising children using different classification decisions (General Point 1). This did not materially affect the findings. We have revised the manuscript to describe this sensitivity analysis in the methods (page 9 and 10) and report the negligible effect in the results (page 15) and discussion section (page 22) but have not presented the results in the revised manuscript.

Results

1. Of the 368 children in the treatment group, it reads that all children were prescribed spectacles? Were there any children that were false positives from the initial screenings – and any emmetropes, i.e. less than 1.00D?

253/368 children in the treatment group had been prescribed spectacles, 112 had failed to attend and no results were available for 3 children. Both reviewers asked about the identification of false positives in the screening programme we have responded to this in General Point 4.

2. On page 12, it is interesting that 20% of children were still wearing glasses in Year 3 but were classified in the non-adherent group (based on not having glasses at the Year 2 follow-up). In my clinical experience, it would seem that the child must be somewhat adherent to spectacle wear if they are still presenting with spectacles two years following the initial examination (or they have been seen elsewhere in the meantime). This is the group that may benefit from re-analysis, so just the 0/2 and 2/2 compliance groups are included.

We have now performed additional analyses as per General Point 1 and point 6 above.

3. Table 2 – it is an interesting finding that the VAs for both better and worse eyes are different between in the adherent and non-adherent groups, yet there is no difference in SER between groups. Can the authors provide any comment as to what they think might be driving this?

Table 2. There is a difference of approximately 2 letters in mean VA, with children in the adherent group having statistically significant lower VA; this difference however, would not be clinically significant. There is no difference between the mean SER measures. We believe this is likely to be the result of the way SER is calculated (sphere plus half cylinder). The mean SER is not statistically different between the groups but there are differences in the proportion of children with astigmatism in the adherent group and non-adherent group which may lead to the visual acuity difference but produce the same mean SER.

4. The results around VA improvement between comparison and treatment groups, and worse and better eyes are a little confusing. They are presented 3 times (abstract, in results with Table 3 and discussion). After reading the abstract, my initial interpretation was that the VA in the better eye had biggest gains in the adherence group, compared with the worse eye, as well as with the non-adherent and comparison group. Then reading the results section, my interpretation changed to the worse eye having the most gains. The summary of this section in the discussion, was the easiest to understand. If the authors could potentially find a more

simple way of presenting this group of results it would make the interpretation a lot easier.

We have revised the text in the abstract and in the results section (pages 14 and 15) in an attempt to more clearly present our findings. We now state the total effect for each group i.e. the effect of age on visual acuity for all children, plus the additional effect of being in either the adherent or non-adherent group.

The revised text (pages 14 and 15) reads;

“The VA of all children improved with increasing age, -0.009 log units per month (95% CI: -0.011 to -0.007) (worse eye) (Table 3).

Over and above this improvement the adherent group (worse eye) improved by a further -0.008 log units per month (95% CI: -0.009 to -0.007). The adherent children therefore improved overall by -0.017 (95% CI -0.020 to -0.015) log units per month (95% CI: -0.009 to -0.007) (approximately two letters every 3 months) and also demonstrated a small amount of improvement in the better eye above that expected from age (Table 3).

The non-adherent group (worse eye) improved by -0.003 log units per month (95% CI: -0.004 to -0.001) above that expected from age. The non-adherent children therefore improved overall by -0.012 log units per month (95% CI: -0.014 to -0.010). No additional improvement above that expected from age was demonstrated in the better eye (Table 3).”

5. In the literacy section, there was no adjustment for SER as it was not associated with letter-ID. Was any separate analysis completed looking specifically at hyperopia and letter-ID, or hyperopia and reduced NVA and letter-ID (VIP-HIP). One might expect differences in association between SER and letter-ID in the hyperope/reduced NVA group, compared with a myopic/reduced DVA group (as per the Dirani paper cited later in the discussion)

We have further analysed the data using refractive error defined by category (General Point 3). The additional analysis demonstrates that no one specific refractive category is associated with letter-ID score. We believe the lack of association we report between SER and literacy is due to the inclusion in our study of a wide range of refractive groups, and believe our findings are related to a number of factors:

- Our study includes a SER ranging from -8.25 to +7.50D and VA's ranging from 0.0 to 1.0 logMAR. We suspect that reduced VA in the presence of refractive error affects literacy. Therefore reduced VA in the presence of hypermetropia or astigmatism will affect the literacy score. In this study we have few cases of myopia but have been able to include all children in the analysis by calculating the SER.
- When we repeat the analyses using refractive categories the numbers of children within each category is insufficient for the analysis to have the statistical power to find an effect as statistically significant.
- Previous cross-sectional studies have reported on one type of refractive error. In the VIP-HIP study participants with hypermetropia ($\geq +3.00$ DS and $\leq +6.00$ DS) were additionally categorised by the level of near VA ($\geq 20/40$ or $< 20/40$). The VIP-HIP study reports hypermetropia in conjunction with near VA less than 20/40 to be associated with reduced literacy. Children with hypermetropia ($\geq +3.00$ DS and $\leq +6.00$ DS) and near VA better than 20/40 did not demonstrate an association with reduced literacy. In addition the VIP-HIP study state that the “Analysis of distance VA resulted in the same qualitative conclusions (data not shown).” It is possible therefore that it is the VA rather than the refractive category that is driving the VIP-HIP study results.
- In the Dirani study of myopic children the authors state that the narrow range of VA (most children had a good level of VA) is a weakness in their study and they therefore cannot conclude definitively that VA is not associated with literacy.

We suggest that our inclusion of all refractive types in our study is a strength, allowing detailed analysis of the association between refractive error, VA and literacy. We thank the reviewer for highlighting these points and we have revised the discussion (page 21 and 22) to include the above points.

Discussion

1. The opening sentence states that this study is the first to assess VA and literacy in children

following screening. There are lots of other studies that have previously investigated this (as the authors discuss later in the discussion). The uniqueness of this study is its attempt to measure adherence/non-adherence to spectacle wear and effect on VA and literacy.

Thank you for highlighting this. We have revised the text accordingly (page 19) to read, "This study is the first longitudinal study to assess the effect of adherence/non-adherence to spectacle wear on VA and literacy in children following vision screening."

2. The authors cover a large number of studies in the discussion. Description of a lot of the studies are minimal, leaving the reader questioning how the findings directly relate to the current study. Perhaps a few of these studies could be removed, and some of the larger studies discussed in more detail. E.g. the large VIP-HIP study currently has a similar amount of explanation as a small pilot study.

We thank the reviewer for their suggestion and have revised the discussion section reducing the number of referenced studies and have provided a more detailed discussion (page 21 and 22).

3. Reference number 34, that uses the addition of +/- 2.00D lenses – it is not clear to me what this means, so needs more explanation or removing.

In view of this comment and that of reviewer 2 we have revised the discussion and removed the reference (34) which cited this test.

4. The sentence on reference number 45 also is not clear, so needs more explanation or removing.

We have now revised the discussion and this reference has been removed.

5. In the 3rd last paragraph, the term moderate hypermetropia is used to describe the range of refractive error in the current study. This is the first time 'moderate' has been used to classify the RE range; in the results low hypermetropia and hypermetropia (>+3.00 are used).

The text has been revised (page 22) to read, "We include children with a wide range of refractive error and VA's allowing a robust analysis of the influence of both factors on developing literacy."

6. Given the large body of evidence linking uncorrected hypermetropia and reading performance, I think the lack of association between SER and literacy in the current study needs covering more in the discussion.

Thank you for highlighting this point. We have now revised the discussion accordingly (page 21).

7. In the second last paragraph, it says DVA is the sole measure of visual function, but NVA is included in the methods sections at follow-up 2 and 3.

In view of comments from both reviewers we have analysed the near VA data collected at time point 3 (General point 2) to examine the correlation between the near VA and distance VA measures.

Reviewer: 2

Page 7: Line 9.

Visual acuity screening in preschool children is known to produce significant false negative and false positive results. How does one know if the children who passed the vision screening did not, in fact

have a significant refractive error? How many children who failed the screening did not have a significant refractive error?

The identification of false positives and false negatives were questioned by both reviewers and replied to in General point 4 above.

Line 29. Can you include a figure of the LogMAR eye chart used in this study so readers internationally know exactly how it is displayed and formatted?

A figure of the logMAR (Keeler) test is now included in the supplementary information.

Line 39. Which model of Welch-Allyn auto refractor was used? Was this the Welch-Allyn "Spot" device?

The Welch Allyn SureSight auto refractor was used in the vision screening regime, the manuscript has now been amended to include this (revised manuscript page 7 and page 9).

Line 45. Spell out NSC. Can you show these criteria in a table?

National Screening Committee is now spelt out in full in the text (page 7). The National Screening Committee guidance is detailed and would not easily assimilate into a table. The reference therefore includes a link to the recommendations should the reader require further detailed information regarding UK vision screening recommendations.

<http://www.screening.nhs.uk/vision-child>

Line 53. How many ophthalmologists and optometrists participated in this study? Was there a significant difference in the spectacle prescribing habits of optometrists vs ophthalmologists?

This is a pragmatic study based on current screening recommendations and clinical practice. There are two ophthalmologists and two optometrists based in the hospital eye service who performed the children's cycloplegic examinations. Those children referred to the community optometrists had a choice of optometrist across the city; in line with current clinical practice we did not stipulate which optometrist children should visit and we did not examine the prescribing differences between the optometrists and the ophthalmologists.

Page 9: Line 15. Can you include a figure of the Bailey-Lovie near vision chart, again for international readers who may be unfamiliar with it?

We have now included a figure of the near vision test in the supplementary information.

Line 41: I am uncomfortable with the definition here of adherence being spectacle wear at the time of follow up assessment. What about all other days up to this assessment?

We were unable to observe the children on a daily basis. The visits to the school for the assessments were unannounced and we believe that the categorisation is generally representative of the children's adherence. This is reflected in the fact that only 39/368 (10.6%) children had different adherence at

the two follow up observations. As part of a separate qualitative study we did interview a small sub-group of parents (this paper is currently in review), which confirmed our categorisation. The categorisation of adherence and non-adherence was questioned by both reviewers and we have, therefore, performed additional sensitivity analysis (pages 9 and 10) varying the categorisation, (General Point 1) with no material difference to the results (page 15).

Page 11: Line 13. Delete the word for “attended the initial cycloplegic exam. . . “

“for” has been deleted.

Lines 15-19: I would also exclude from this study those children who had incomplete assessments; i.e. 1) no cycloplegic refraction, 2) No follow up visual acuity, 3) failed to attend follow up appts. Or justify their inclusion.

As the study was pragmatic and we were observing the children’s attendance and adherence, we were unable to perform cycloplegic refraction for children if they failed to attend. To address the issue of missing data the statistical model selected for the analyses, using projections over time, takes into account missing data and requires a minimum of measures at two time points. Using this type of statistical analysis allows inclusion of a greater number of participants giving maximum power to the analyses. We have now stated in the methodology section that the choice of model used for the analysis takes into account missing data (page 9).

Line 27: The definition for myopia starting at only 0.5 DS sphere is rather mild, especially if one is looking at literacy related to refractive error. Children who have only 0.5 DS will invariably still have normal vision for reading and other near work, even if they do not have glasses.

The definition for myopia was chosen to be comparable with the refractive error categorical definitions used with young children in other studies.^{3,4} We agree with the reviewer that children with low degrees of myopic refractive error are likely to have a normal level of near visual acuity even without their spectacles. We have examined the data using the myopia category as ≤ -1.00 DS, there are only three children that would be reclassified as myopic, this does not materially change the results and we have therefore retained our original classification.

References:

3. Robaei D, Rose KA, Ojaimi E, et al. Causes and associations of amblyopia in a population-based sample of 6-year-old Australian children. *Arch Ophthalmol* 2006;124(6):878-84.
4. O'Donoghue L, McClelland JF, Logan NS, et al Refractive error and visual impairment in school children in Northern Ireland. *British Journal of Ophthalmology* 2010;94:1155-1159.

Line 37. Occlusion therapy is usually prescribed for amblyopia, not strabismus. If the 14 children with strabismus, also had amblyopia, this should be stated.

Seven of the 14 children with strabismus had occlusion therapy confirmed in their medical notes of these five had amblyopia. This is now stated in the text (revised manuscript page 12) “Fourteen of the 368 (3.8%) children had a constant or intermittent strabismus, five of whom had been prescribed occlusion therapy for amblyopia. Those children were not excluded from the analysis as they met the initial VA referral criteria and had been prescribed spectacles.”

Page 18:Line 33. Change the word reduces to declines (... by approximately 1.5%. . .)

Reduced has now been changed to “declines” (revised manuscript pages 17 and 19).

Page 19: Line 17. The +2.00D lens test has never been validated as a useful or effective test for hyperopia.

We thank the reviewer for highlighting this. In addition we take on board the comment of reviewer 1 (Discussion point 3) we have, therefore, revised the discussion and removed the reference (34) citing this test.

Page 21: Line 13. I agree that near acuity testing, as opposed to or in addition to distance acuity testing, would have been a very nice (and perhaps very important) variable to add to this study.

In view of comments from both reviewers (General Point 2), we now report the correlation between the near and distance VA at time point 3 and include a supplementary table with the results. The discussion in the revised manuscript (page 21 and 22) has been altered to reflect these findings.

VERSION 2 – REVIEW

REVIEWER	Shelley Hopkins QUT, Australia
REVIEW RETURNED	23-Mar-2018

GENERAL COMMENTS	Second review of: Effect of adherence to spectacle wear on early developing literacy: a longitudinal study based in a large multi-ethnic city, Bradford, UK. General comments: the authors have made some significant improvements to the manuscript in their revision. Upon second review, I still have some questions around methodology – which is still requiring more clarity. Introduction: Second sentence – end of sentence, reduction in VA should be changed to ‘potential reduction in VA’ given it is based on a screening outcome. Third sentence – Change start of sentence to... ‘For those children who fail the screening...’ End of sentence could be changed to ‘confirm the VA finding and to determine the presence and magnitude of any ...’ Fourth sentence – Surgery should be added as the final treatment option, i.e. spectacles, occlusion or surgery. Last sentence – needs rewording... on early developing VA and literacy skills should be ... on VA and early developing literacy skills... Methods:
---

	Line 6: replace 'linked' with 'evaluated' Population – last sentence: The cohort is broadly representative of the city's maternal population. What does this mean? Of Bradford's mothers – all ages?? Patient and Public Involvement: I find this whole section unusual to include in the manuscript. It also isn't clear if the BiB project team, eluded to in the opening sentence, refers to the authors of the current manuscript, or are a different team responsible for the larger epidemiological study. Recruitment – opening sentence: It is not clear who the 'Starting Schools Programme' fits in with the BiB group. Were all children recruited in the current study from the Starting Schools Programme? Figure 1 reads that 944 children participated, and only 432 were from the Starting Schools Programme? Baseline Vision Assessments – Year 1, para 2, sentence 1. So only children with reduced VA and strab fail screening? How are the results of the ocular motility assessment and Welch-Allyn SureSight screener incorporated into the pass/fail criterion then, as they are also performed in the vision screenings. Sentence 3 – need to define 'low degrees of hypermetropia' as this is first time this term is introduced in manuscript. Baseline Vision Assessments – Year 1, para 3. So this data vision screening and first follow-up (community optom/hospital eye clinic) was collected retrospectively? If so, this needs to be made clear in the opening sentence of the methods where the study is called prospective. Baseline Literacy Assessments – Year 1, last sentence. Can the BPVS be described in more detail? It is not entirely clear why the test is being included – that is, as a stand-alone academic measure? The accuracy of some of the conclusions made later in the manuscript around BPVS (see later comments) is unclear, as little information is provided around this test, and what it is measuring, in particular, in area of cognitive ability – what about children from different language backgrounds. Follow-up Assessments – Years 2 and 3, sentence 2: ...unaware of the previous year's vision or literacy results... in this context, previous year means Year 2, as Year 1 was collected by non-research team. If correct, please clarify in manuscript. Sentence 5: is the cover test, ocular motility, non-cycloplegic auto refraction data used anywhere in the analysis? The use of the SureSight Vision screener seems a bit redundant in this study? Statistical Analysis – Analysis of Literacy, sentence 3: list the
--	---

	factors specifically that were used in the analysis that have been associated with educational attainment Sentence 4: BPVS score was used to account for language ability – but later, it is concluded that it is related to cognitive ability? More about this test needs explaining. The authors should explain why they have controlled for language ability when looking at early literacy, but later said that this controls for cognitive ability? How does the test differentiate between children who are from other language backgrounds, and cognitive ability? Statistical Analysis – Visual Acuity Time Point 3 Is time point 3, Year 3? If so, please change to this. Years 1, 2 and 3 have been used throughout manuscript except for this section discussing Time Point 3. Results I think the first and second sentences of the opening paragraph should be swapped around. Rather than beginning on participants excluded, it presents better to have the whole numbers first. I.e. ‘Data from 801/944 (85%) children from ...’ Mid-way through first para: the grouping of the refractive errors places 1.00DC astigmatism as the first group (alone or in combination with hyperopia and myopia), followed by hyperopia and myopia alone. Does this mean that there are some hyperopes and myopes with astigmatism in the astigmatism group that aren’t presented in the hyperope/myope numbers? Perhaps a better way of presenting this would be to present all REs as alone or in combination with astigmatism, and recognise that there will be overlap. As it stands, where are the hyperopes >3.00D with astigmatism? Last sentence of first paragraph: have the authors re-run the analysis without the amblyopes in the analysis. And also, are the amblyopes in the adherent or non-adherent group. If the amblyopes have had history of treatment plus occlusion therapy, and still failed the screening (met criteria for study), it would be assumed that their visual profile, and expected improvements following spectacle wear will be different to those without amblyopia. This may gains made in VA may be expected to be more if these children were excluded? Second para: VA’s should be VAs Sentence before Table 2: ... suggesting that there were no differences in cognitive ability. Is this an accurate statement? Results – Visual Acuity: what is mean diff? (mean diff: 0.337logMAR) and later (mean diff: 0.273 logMAR) Results – Visual Acuity, second para: The VA of all children improved with increasing age... why is only worse eye reported here. Best eye results should also be reported here and in next
--	---

	paragraph. Also, what happens to these results if amblyopes are taken out of analysis? Results – Visual Acuity at Time Point three – same comments as earlier around changing this to Year 3, if accurate. Discussion A sentence needs to be included between the first and second sentence of the opening para to comment on results around VA and spec adherence (before discussion of relationship with early literacy). Fourth para: ‘In a Singaporean study, a strong association between paternal level of education and ...’ Is this correct – Paternal? The rest of para relates to maternal. Seventh para: ‘Classroom based tasks where fixation frequently changes...’ This should be a new paragraph, not at the end of the astigmatism para. Eighth para: VA’s should be VAs Tenth para: Visual acuity is the sole measure of visual function... In this para, the authors should consider the limitation of using a letter visual acuity test (albeit with matching card) and letter-ID recognition as the two assessment tests. It would seem that children with poor letter ID would also struggle with letter VA based on the nature of the test, and may demonstrate lower confidence in completing the test even with the assistance of the matching card. Final para: ‘The children who do not adhere to spectacle wear are likely to be those in families who are poorer and less educated’. This isn’t what the study’s results say. In Table 2, only maternal education level is related to spec adherence. This needs to be tighter.
--	---

REVIEWER	Geoff Bradford, MD West Virginia University, United States
REVIEW RETURNED	29-Mar-2018

GENERAL COMMENTS	I appreciate the revisions the authors have endeavored to include from my initial review and feel this manuscript can now be accepted for publication.
--

VERSION 2 – AUTHOR RESPONSE

We thank the reviewer for their comments on the revised manuscript. We have responded to the points raised on a point by point basis and have revised the manuscript accordingly.

Introduction:

Second sentence – end of sentence, reduction in VA should be changed to ‘potential reduction in VA’ given it is based on a screening outcome.

The sentence has been revised as requested.

Third sentence – Change start of sentence to... ‘For those children who fail the screening...’ End of sentence could be changed to ‘confirm the VA finding and to determine the presence and magnitude of any ...’

The sentence has been revised as requested.

Fourth sentence – Surgery should be added as the final treatment option, i.e. spectacles, occlusion or surgery.

The treatment stated relates to the visual acuity therefore surgery has not been included in the sentence.

Last sentence – needs rewording... on early developing VA and literacy skills should be ... on VA and early developing literacy skills...

The sentence has been revised as requested.

Methods:

Line 6: replace ‘linked’ with ‘evaluated’

The study uses data from healthcare, education and research which was linked. The sentence has been revised to clarify this and now reads, “Baseline epidemiological data collected from mothers and children of the BiB cohort, literacy measures, vision screening results and repeat measures of vision and literacy were linked in order to evaluate the longitudinal impact of adherence to spectacle wear on VA and early literacy.”

Population – last sentence: The cohort is broadly representative of the city’s maternal population.

What does this mean? Of Bradford's mothers – all ages??

The sentence has been revised to read, "The cohort is broadly representative of the city's maternal population of child bearing age."

Patient and Public Involvement:

I find this whole section unusual to include in the manuscript. It also isn't clear if the BiB project team, eluded to in the opening sentence, refers to the authors of the current manuscript, or are a different team responsible for the larger epidemiological study.

The section describing the population and the Patient and Public Involvement section have been included as a mandatory request by the journal. The opening sentence refers to the BiB project and this has now been clarified in the text.

Recruitment – opening sentence:

It is not clear who the 'Starting Schools Programme' fits in with the BiB group. Were all children recruited in the current study from the Starting Schools Programme? Figure 1 reads that 944 children participated, and only 432 were from the Starting Schools Programme?

The opening sentence has been revised and now reads, "As part of the BiB study children's literacy levels on school entry (termed 'Reception Class' in England, UK and defined as Year 1 of this study) were measured between September 2012 and July 2014 in Bradford schools. 2930 BiB children from seventy-four of the one hundred and twenty-three primary schools (60%) participated. 432 of the 2930 (14.7%) failed their vision screening (Figure 1) and were referred for follow-up cycloplegic investigation, these children are defined as the treatment group." The legend for Figure 1 has been amended to clarify the recruitment as stated above.

Baseline Vision Assessments – Year 1, para 2, sentence 1. So only children with reduced VA and strab fail screening? How are the results of the ocular motility assessment and Welch-Allyn SureSight screener incorporated into the pass/fail criterion then, as they are also performed in the vision screenings.

Only children with reduced VA or strabismus failed the vision screening. The additional tests were performed as part of the screening process however the results of these tests are being used as part of a research paper evaluating vision screening tests and the data are not used in the analyses in this

paper. In order to ensure clarity within the paper references to ocular motility and Welch-Allyn SureSight have now been removed.

Sentence 3 – need to define ‘low degrees of hypermetropia’ as this is first time this term is introduced in manuscript.

The definition has now been included.

Baseline Vision Assessments – Year 1, para 3. So this data vision screening and first follow-up (community optom/hospital eye clinic) was collected retrospectively? If so, this needs to be made clear in the opening sentence of the methods where the study is called prospective.

The study design is prospective with the children identified at the point of screening (Year 1) and VA and literacy measures repeated annually. However, the data from the medical notes which was used to confirm the presence of refractive error (cycloplegic examination), the presence or absence of pathology or strabismus could not be extracted until after the children had received their cycloplegic assessment.

Baseline Literacy Assessments – Year 1, last sentence. Can the BPVS be described in more detail? It is not entirely clear why the test is being included – that is, as a stand-alone academic measure? The accuracy of some of the conclusions made later in the manuscript around BPVS (see later comments) is unclear, as little information is provided around this test, and what it is measuring, in particular, in area of cognitive ability – what about children from different language backgrounds.

The BPVS test is a measure of receptive vocabulary (ability to orally identify pictures) and is a proxy measure for cognitive ability in young children. It has been included in the analysis to account for any difference in cognitive ability between the groups. The text has been revised and now reads, “In addition receptive vocabulary was measured using the British Picture Vocabulary Scale (BPVS), providing a representation of IQ in young children. This measure is included to adjust for potential confounding due to levels of general cognitive ability.”

Follow-up Assessments – Years 2 and 3, sentence 2: ...unaware of the previous year’s vision or literacy results... in this context, previous year means Year 2, as Year 1 was collected by non-research team. If correct, please clarify in manuscript.

The sentence has been revised, “Both the vision and the literacy assessments were administered on the same day by the same personnel who were unaware of previous vision or literacy results.”

Sentence 5: is the cover test, ocular motility, non-cycloplegic auto refraction data used anywhere in the analysis? The use of the SureSight Vision screener seems a bit redundant in this study?

As per the point above reference to additional tests have been removed from the text.

Statistical Analysis – Analysis of Literacy, sentence 3: list the factors specifically that were used in the analysis that have been associated with educational attainment

The individual factors are listed in the previous paragraph; the text has been amended to state “as above”.

Sentence 4: BPVS score was used to account for language ability – but later, it is concluded that it is related to cognitive ability? More about this test needs explaining. The authors should explain why they have controlled for language ability when looking at early literacy, but later said that this controls for cognitive ability? How does the test differentiate between children who are from other language backgrounds, and cognitive ability?

As stated above (Baseline Literacy Assessments – Year 1, last sentence) the inclusion of the BPVS has now been described in detail.

Statistical Analysis – Visual Acuity Time Point 3

Is time point 3, Year 3? If so, please change to this. Years 1, 2 and 3 have been used throughout manuscript except for this section discussing Time Point 3.

This has now been changed from time point 3 to Year 3.

Results

I think the first and second sentences of the opening paragraph should be swapped around. Rather than beginning on participants excluded, it presents better to have the whole numbers first. I.e.

'Data from 801/944 (85%) children from ...'

The sentences have been swapped as requested.

Mid-way through first para: the grouping of the refractive errors places 1.00DC astigmatism as the first group (alone or in combination with hyperopia and myopia), followed by hyperopia and myopia alone. Does this mean that there are some hyperopes and myopes with astigmatism in the astigmatism group that aren't presented in the hyperope/myope numbers? Perhaps a better way of presenting this would be to present all REs as alone or in combination with astigmatism, and recognise that there will be overlap. As it stands, where are the hyperopes >3.00D with astigmatism?

We now present the subdivision of refractive types within the astigmatic category providing the exact number of children with each type of refractive error. "Of the 253 children in the treatment group with cycloplegic refraction results, 157/253 (62.1%) had astigmatism (>1.00DC) either alone (n=19) or in combination with hypermetropia (>+3.0DS) (n=56), low hypermetropia (>+1.0DS to +3.0DS) (n=16) or myopia (\leq -0.50DS) (n=66). 35/253 (13.8%) had hypermetropia alone, 11 (4.3%) had myopia alone and 50 (19.8%) children had low hypermetropia. 55 of 253 (21.7%) additionally had anisometropia (\geq 1.0D difference)."

Last sentence of first paragraph: have the authors re-run the analysis without the amblyopes in the analysis. And also, are the amblyopes in the adherent or non-adherent group. If the amblyopes have had history of treatment plus occlusion therapy, and still failed the screening (met criteria for study), it would be assumed that their visual profile, and expected improvements following spectacle wear will be different to those without amblyopia. This may gains made in VA may be expected to be more if these children were excluded?

We have re-run the analysis excluding the amblyopic children, no material difference was found in the results.

Second para: VA's should be Vas

The text has been amended accordingly.

Sentence before Table 2: ... suggesting that there were no differences in cognitive ability. Is this an accurate statement?

The sentence has been revised to read, "BPVS did not differ between the adherent and non-adherent children ($p=0.553$) suggesting that there were no differences in cognitive ability."

Results – Visual Acuity: what is mean diff? (mean diff: 0.337logMAR) and later (mean diff: 0.273 logMAR)

Mean diff is the mean difference. The sentence has been revised for greater clarity, "At baseline compared to the comparison group, both the adherent (mean difference: 0.337 logMAR; 95% CI: 0.304 to 0.370) and non-adherent groups (mean difference: 0.273 logMAR; 95% CI: 0.241 to 0.305) had lower levels of VA in the worse eye."

Results – Visual Acuity, second para: The VA of all children improved with increasing age... why is only worse eye reported here. Best eye results should also be reported here and in next paragraph. Also, what happens to these results if amblyopes are taken out of analysis?

The results of the better eye were removed from the original manuscript to simplify the text, the results for both eyes are now included in the text and in Table 3.

Results – Visual Acuity at Time Point three – same comments as earlier around changing this to Year 3, if accurate.

This has now been changed from time point 3 to Year 3.

Discussion

A sentence needs to be included between the first and second sentence of the opening para to

comment on results around VA and spec adherence (before discussion of relationship with early literacy).

Additional text has now been included, "The VA of children who adhered to spectacle wear was found to improve at a far greater rate compared to those who were non-adherent, with the VA of adherent children reaching similar levels to the VA of the comparison children by the end of the study."

Fourth para: 'In a Singaporean study, a strong association between paternal level of education and ...' Is this correct – Paternal? The rest of para relates to maternal.

This is a correct statement, the Singapore team use paternal education as their indicator of parental education.

Seventh para: 'Classroom based tasks where fixation frequently changes...' This should be a new paragraph, not at the end of the astigmatism para.

New paragraph inserted.

Eighth para: VA's should be Vas

The text has been amended accordingly.

Tenth para: Visual acuity is the sole measure of visual function... In this para, the authors should consider the limitation of using a letter visual acuity test (albeit with matching card) and letter-ID recognition as the two assessment tests. It would seem that children with poor letter ID would also struggle with letter VA based on the nature of the test, and may demonstrate lower confidence in completing the test even with the assistance of the matching card.

The limitations of using a letter test for both VA and literacy is considered and the text has been revised to read, "The VA assessment and the literacy test are both letter based and children who struggle with letter identification may also demonstrate a poor ability with the VA test. However, all children used a matching technique, a skill that is present in children as young as three years⁴⁶ and no child who failed the screening was classed as false positive."

Final para: 'The children who do not adhere to spectacle wear are likely to be those in families who are poorer and less educated'. This isn't what the study's results say. In Table 2, only maternal education level is related to spec adherence. This needs to be tighter.

The sentence has been amended and "poorer" has been removed from the text.